# Proteus: Lookup-Free Trellis-Coded Quantization by Lattice-Breaking Compute Codes for 2-Bit LLMs

**Zhengwu Yang**[1] **Xunchao Li**[1] **Ke Cheng**[1] **Kunlong Liu**[1] **Jianfeng Yang**[1] **Haoshuang Wang**[1] **Kaipeng Deng**[1] **Qingqing Dang**[1] **Yanlin Sha**[1] **Yanjun Ma**[1] **Dianhai Yu**[1]

## Abstract

Autoregressive decoding of large language models is frequently memory-traffic bound, so ultra-low-bit weight-only PTQ helps only if dequantization avoids irregular codebook or LUT access in the inner loop. Under the GPU-friendly bitshift trellis, existing 2-bit trellis-coded quantization (TCQ) pipelines either reintroduce micro-LUTs or suffer overlap-amplified artifacts because incoherence improves global Gaussianity but does not guarantee overlap-local joint geometry. We introduce Proteus[1], a strictly lookup-free TCQ framework whose computed generator MUL-BAL uses cheap integer mixing plus a per-layer affine Gaussianizer to produce overlap-robust, near-Gaussian code values with zero runtime table loads. Proteus instantiates each layer by selecting from a tiny, pre-vetted candidate pool and then applies lightweight channel compensation and optional few-shot distillation that tune only per-layer affine statistics while keeping packed indices and the bitshift-trellis decoder fixed. On Llama 2 (7B–70B) at 2-bit PTQ, Proteus improves perplexity and zero-shot accuracy over strong TCQ/PTQ baselines and reduces end-to-end decode bandwidth at comparable throughput (e.g., 740 vs. 1020 GB/s on 70B).

## 1. Introduction

Large language models (LLMs) are increasingly served in latency-sensitive autoregressive decoding, where end-to-end throughput is often constrained by *memory traffic*—moving model weights and managing the KV cache—rather than

peak FLOPs (Kwon et al., 2023). Weight-only post-training quantization remains a primary lever: reducing weight precision lowers bytes moved per generated token, but the benefit materializes only if dequantization remains *deployment-friendly* and does not reintroduce irregular memory-side bottlenecks (Xiao et al., 2023; Frantar et al., 2022).

**Gaussianized PTQ and the rise of ultra-low-bit structure.** Recent ultra-low-bit PTQ pipelines increasingly follow a "Gaussianized" recipe: apply an *incoherence* transform (e.g., randomized orthogonal/Hadamard rotations) so that transformed weights behave closer to an i.i.d. Gaussian reference, and then solve a second-order rounding problem under a quadratic proxy (Chee et al., 2023; Tseng et al., 2024a; Frantar et al., 2022). This recipe has unlocked viable compression in the extreme regime (2–4 bits/weight), yet it also exposes a tension between *expressivity* (vector codebooks / structured rounding) and *deployability* (tight kernel budgets and predictable memory access).

**Why trellis-coded quantization (TCQ) is attractive—and where it breaks under deployment.** Within this landscape, *trellis-coded quantization* (TCQ) is particularly compelling because it achieves high-dimensional vector quantization at extreme bit-rates via dynamic programming, without requiring an exponentially large codebook (Fischer et al., 1991). QTIP operationalizes TCQ for LLM PTQ using a GPU-friendly *bitshift trellis*, where each decoded scalar is produced from an overlapping sliding window of the compressed bitstream (Tseng et al., 2024b). Crucially, the bitshift trellis couples consecutive outputs through heavy window overlap: along each trellis edge, the top bits of the next state must match the bottom bits of the current state (a structure originally introduced in Mao & Gray's random permutation trellis coder) (Tseng et al., 2024b). This overlap enables fast parallel decoding, but it also imposes a stringent *joint-geometry* constraint: the generator is repeatedly queried on highly correlated windows.

**Two deployment-facing gaps.** Despite the promise of TCQ, two bottlenecks remain unresolved in the ultra-low-bit regime.

**(1) Lookup resurgence in high-fidelity deployments.** While "compute codes" motivate lookup-free decoding,

---

[1]Baidu Inc., Beijing, China. Correspondence to: Yanjun Ma <mayanjun02@baidu.com>, Dianhai Yu <yudianhai@baidu.com>.

*Proceedings of the $43^{rd}$ International Conference on Machine Learning*, Seoul, South Korea. PMLR 306, 2026. Copyright 2026 by the author(s).

[1]https://github.com/YZW-explorer/Proteus

strong realizations under tight kernel budgets frequently rely on small LUT or micro-LUT hybrids to realize more expressive codebooks. Even when such tables are tiny (e.g., cache-resident hybrids), they reintroduce irregular on-chip memory accesses into the dequantization inner loop and complicate predictable scaling at extreme throughput (Tseng et al., 2024b; Egiazarian et al., 2024).

**(2) An assumption gap between global Gaussianization and overlap-local trellis geometry.** Incoherence improves Gaussianity in aggregate, but the trellis solver/decoder operates on *local* windows and couples consecutive outputs through overlap. As a result, local tiles can deviate visibly from an i.i.d. Gaussian reference even after incoherence, and mild anisotropies in a generator can be amplified into structured *neighbor artifacts* in the reachable set $\{(y_t, y_{t+1})\}$ under bitshift overlap. These artifacts reduce effective joint coverage under the trellis constraint, precisely where ultra-low-bit TCQ cannot afford wasted codebook capacity.

**Our approach: PROTEUS.** We present PROTEUS, an end-to-end, *strictly lookup-free* TCQ framework for ultra-low-bit (e.g., 2-bit) LLM PTQ. PROTEUS replaces stored codebooks with *computed*, overlap-robust generators and explicitly targets the blockwise geometry induced by bitshift-trellis decoding. At its core is a compute-only generator family (**MUL-BAL**) that combines lightweight integer mixing with a stable Gaussianizer, designed to disrupt overlap-amplified regularities while preserving near-Gaussian marginals. To keep deployment friction minimal, PROTEUS instantiates each layer by selecting the discrete generator from a *tiny, pre-vetted candidate pool* (screened offline for spectral quality and overlap robustness), and then applies only lightweight continuous compensation. Concretely, PROTEUS decouples compression into three stages: (i) a cheap, layerwise discrete search (Hessian-weighted proxy) that selects a layer-specific generator and packs trellis indices; (ii) lightweight per-layer continuous compensation (e.g., channel-wise scales) with the discrete structure frozen; and (iii) an optional global few-shot relaxation that tunes only a tiny set of per-layer affine statistics while keeping indices and the decoder unchanged. This preserves hard deployment invariants—fixed bitshift-trellis decoding and *zero* runtime codebook loads—while improving the local isotropy that the solver actually sees.

**Contributions.** PROTEUS (i) enables strictly lookup-free TCQ under the bitshift trellis while retaining trellis rate–distortion benefits (Tseng et al., 2024b; Fischer et al., 1991); (ii) introduces **MUL-BAL**, an overlap-robust computed Gaussian generator that improves reachable-set geometry beyond marginal Gaussianity; (iii) instantiates layer-specific generators via a cheap Hessian-weighted discrete proxy over a tiny, pre-vetted candidate pool (Frantar et al., 2022; Chee et al., 2023); and (iv) decouples discrete structure selection

from minimal continuous compensation and optional few-shot distillation, keeping the decoder and packed indices fixed at deployment.

## 2. Background and Motivation

Our goal is to motivate why ultra-low-bit PTQ increasingly gravitates toward *Gaussianized* structured quantizers, and why TCQ under a bitshift trellis introduces two deployment-critical gaps that PROTEUS targets: (i) *inner-loop sensitivity to lookup/codebook traffic*, and (ii) a *global-to-local assumption gap* where aggregate Gaussianization does not guarantee overlap-robust local geometry.

**Related work (deployable PTQ).** Deployable LLM PTQ typically starts from scalar/groupwise kernels and recovers accuracy via calibration/range control (Zhao et al., 2025a; Xiao et al., 2023; Lin et al., 2024; Shao et al., 2024; Edalati et al., 2025; Ghaffari et al., 2025), outlier/heteroscedasticity handling (Dettmers et al., 2022; Kim et al., 2024; Dettmers et al., 2024; Yu et al., 2024; Liang et al., 2025; Qiao & Huang, 2026), layer/block reconstruction (Frantar & Alistarh, 2022; Frantar et al., 2022; Lee et al., 2025; Song et al., 2025), and format/system co-design (Dettmers et al., 2023; 2024; Zhu et al., 2025; Lee & Song, 2025; Huang et al., 2025); benchmarks also note brittleness under distribution/prompt shifts (Chang et al., 2025).

**Gaussianized PTQ.** At 2–4 bits, naive rounding can sharply degrade perplexity (Wei et al., 2022); *Gaussianized* PTQ instead applies incoherence (e.g., Hadamard/orthogonal rotations) and then performs curvature-aware quadratic-proxy rounding (Liu et al., 2025; Zhao et al., 2025b; Chee et al., 2023; Tseng et al., 2024a; Frantar et al., 2022; Ashkboos et al., 2024; Xiang & Zhang, 2024; Choi et al., 2025; He et al., 2025; Maisonnave et al., 2025).

**Why structured quantizers at 2-bit.** Residual *local geometry* (correlation/heavy tails/structured components) motivates vector/lattice/trellis quantizers beyond scalar/groupwise rules (Chee et al., 2023; Tseng et al., 2024a; Lee & Song, 2025; Malinovskii et al., 2025; Zhang et al., 2025; Egiazarian et al., 2024; Liu et al., 2024; Xu et al., 2025; Yue et al., 2025; Zhou et al., 2025; Tseng et al., 2024b; Lee & Song, 2025), but added expressivity often comes with codebook/lookup friction in the dequantization inner loop.

**TCQ under a bitshift trellis: overlap $\Rightarrow$ joint-geometry constraints.** Trellis-coded quantization (TCQ) is a classical approach to high-dimensional quantization via dynamic programming on a trellis (Fischer et al., 1991). QTIP adopts a hardware-friendly *bitshift trellis* (from the random-permutation trellis-coder line) so each output is generated from an $L$-bit sliding window of a packed stream; consecutive windows shift by $\Delta \triangleq kV$ bits, hence sharing $L - \Delta$

bits (Mao & Gray, 2010; Tseng et al., 2024b). Equivalently, along each trellis edge, the next state must match the current state on the shared-bit suffix/prefix, which makes neighboring inputs highly correlated. This structure removes sequential decoding bottlenecks and enables parallel decoding, but it also imposes a stricter requirement on the generator: it must sufficiently *mix shared bits* so that overlap does not collapse the reachable neighbor set $\{(y_t, y_{t+1})\}$ onto low-dimensional slices.

**Gap 1: lookup/codebook traffic resurges in the inner loop.** A naive TCQ deployment requires storing a large node-value codebook (and, depending on the trellis, additional structure), which is incompatible with fast inference at the $L$ needed for high quality. QTIP addresses this by combining incoherence processing, a bitshift trellis (to avoid storing trellis structure), and compute-based Gaussian codes (to avoid storing the full codebook) (Tseng et al., 2024b). Yet, more broadly, high-fidelity deployments under tight kernel budgets still often introduce LUT/micro-LUT hybrids (or other auxiliary structures) to match instruction sets or cache hierarchies, reintroducing irregular memory behavior in the dequantization inner loop (Tseng et al., 2024b; Egiazarian et al., 2024; Dettmers et al., 2024). This motivates a strict design constraint for PROTEUS: *no inference-time codebook/table loads in the decoding inner loop*.

**Gap 2: global Gaussianization does not certify overlap-robust local geometry.** Incoherence improves Gaussianity in aggregate, but the solver/decoder operates on *local tiles/windows*. Empirically, local tiles can deviate visibly from an i.i.d. Gaussian reference even after incoherence (Fig. 1). More importantly, overlap couples consecutive queries: residual anisotropy, correlation, or low-dimensional structure in a generator can be amplified into structured artifacts in $\{(y_t, y_{t+1})\}$ (Fig. 3). Crucially, this failure mode is not reliably detected by one-dimensional marginal tests alone: two generators can have similar marginals yet differ substantially in joint reachable-set coverage.

**Why local mismatch matters under overlap.** Modern "Gaussianized" PTQ applies an incoherence transform (e.g., randomized Hadamard transforms) so the transformed weights look closer to an i.i.d. Gaussian distribution *globally*. However, the downstream solver/decoder is local and overlap-coupled: it operates on blocks and, in the bitshift trellis, consecutive outputs share a large fraction of input bits. Figure 1 visualizes this gap by z-scoring *local tiles* of the incoherence-processed matrix (here $g{=}128$): tiles can deviate visibly from an i.i.d. Gaussian reference, and some deployable code realizations imprint discrete-support / spike artifacts.

**Bitshift-trellis decoding and overlap.** We use the QTIP-style *bitshift trellis*: let the compressed representation be a bitstream (equivalently, $k$-bit symbols $\{z_t\}$) from which

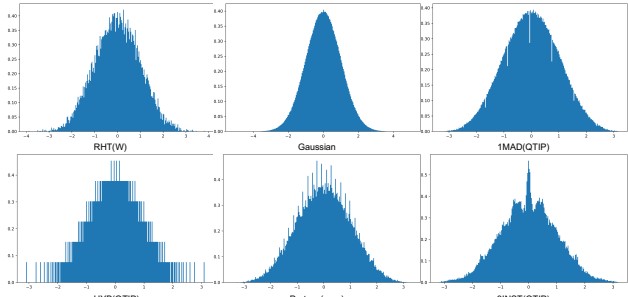

*Figure 1.* **Local-tile distribution mismatch after incoherence.**

we extract an $L$-bit sliding window $x_t \in \{0, \ldots, 2^L - 1\}$ at each step. A fixed compute generator $g(\cdot)$ maps each window to a scalar,

$$y_t = g(x_t), \qquad x_t = \text{WINDOW}(Z; t, L, k), \qquad (1)$$

and the trellis overlap implies $x_{t+1}$ shares $L - k$ bits with $x_t$. Therefore, insufficient mixing inside $g(\cdot)$ can create structured neighbor artifacts in the reachable set $\{(y_t, y_{t+1})\}$ even when the marginal of $y_t$ is close to Gaussian.

**Motivation: lattice-breaking compute codes + cheap layerwise instantiation.** From a generator-design viewpoint, overlap sensitivity is a geometric problem. Congruential constructions inherit lattice/hyperplane structure in low-dimensional projections; the spectral test formalizes this phenomenon by measuring the largest inter-hyperplane spacing, where poor multipliers yield points concentrated on a small number of hyperplanes (Tseng et al., 2024b). To close the gap, PROTEUS adopts three principles: (i) the generator should be *lattice-breaking under overlap*, explicitly targeting joint reachable-set isotropy rather than marginals alone; (ii) generator choice should be *layer-aware*, since layers differ in post-incoherence statistics and curvature sensitivity, motivating lightweight per-layer discrete instantiation from a small fixed candidate set; and (iii) because quadratic proxies and local Gaussianization are imperfect at 2-bit, PROTEUS adds minimal continuous adaptation to absorb residual mismatch *without* changing the inference-time trellis decoder.

## 3. Algorithm

**Overview.** PROTEUS decouples the problem into three stages: *Stage I* selects a *layer-specific* compute generator and solves for trellis indices under a Hessian-weighted proxy; *Stage II* applies lightweight channel-wise compensation with the discrete structure frozen; *Stage III* optionally performs a global few-shot relaxation by tuning only a tiny set of *per-layer* continuous statistics while keeping indices and the decoder unchanged.

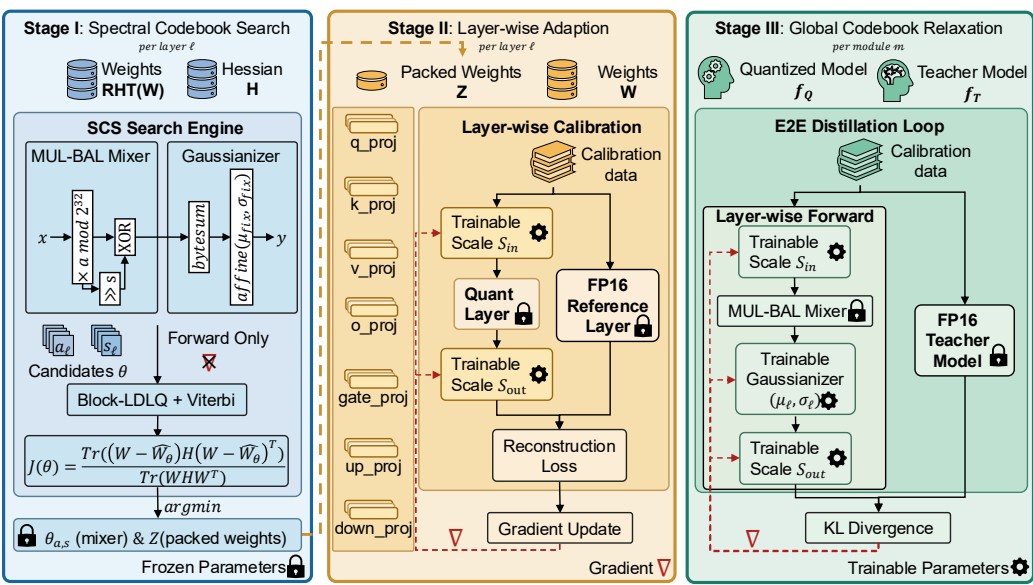

*Figure 2.* **Overview of PROTEUS.**

## 3.1. Compute generator: MUL-BAL

**Context: lookup-free Gaussian codes under bitshift overlap.** In a QTIP-style *bitshift trellis*, each decoded group depends on an $L$-bit window of the packed bitstream, and consecutive windows shift by $\Delta$ bits; in our implementation $\Delta \triangleq kV$ (each step decodes $V$ values, $k$ bits each), which reduces to $\Delta = k$ when $V = 1$. Equivalently, along every trellis edge $(i \to j)$, the top $L - \Delta$ bits of $j$ equal the bottom $L - \Delta$ bits of $i$. This shared-bit constraint is the root cause of *overlap-induced dependence*: even if a generator produces an approximately Gaussian *marginal* $y_t = g(x_t)$, correlated inputs $(x_t, x_{t+1})$ can still collapse the *joint* neighbor set $\{(y_t, y_{t+1})\}$ onto low-dimensional slices (banding, arms, grids). Our design target is thus stricter than "make $y_t$ Gaussian": we want a *compute-only* generator that (i) disrupts overlap-amplified regularities in the joint geometry, (ii) preserves near-Gaussian marginals for Gaussian source coding, and (iii) remains strictly lookup-free in the inner-loop.

**MUL-BAL: a computed Gaussian generator with per-layer affine statistics.** Given an $L$-bit window $x$ (zero-left-padded and interpreted as a 32-bit unsigned integer), MUL-BAL computes

$$\tilde{x}_1 = (a \cdot x) \bmod 2^{32}, \tag{2}$$

$$\tilde{x}_2 = \tilde{x}_1 \oplus (\tilde{x}_1 \gg s), \tag{3}$$

$$u = \sum_{i=0}^{3} \left( (\tilde{x}_2 \gg 8i) \ \& \ 255 \right), \tag{4}$$

$$y = \frac{u - \mu_\ell}{\sigma_\ell}. \tag{5}$$

The name **MUL-BAL** denotes the two operations at its core—the modular *MUL*tiplication in Eq. (2) and the

*BAL*ance step (XOR-with-shift) in Eq. (3)—and refers to the compute generator *inside* PROTEUS; it is a component of our framework rather than a separate, standalone method. The *discrete* parameters $\theta_\ell = (a_\ell, s_\ell)$ are selected in Stage I for each layer $\ell$, while $(\mu_\ell, \sigma_\ell)$ are *continuous, per-layer* affine statistics. We initialize $(\mu_\ell, \sigma_\ell)$ by the i.i.d. byte-uniform heuristic implied by Eq. (4), and optionally refine them in Stage III *independently per layer/module*. This keeps the generator layer-customized without introducing any inference-time codebook loads.

**Design principle: high-to-low feedback for overlap robustness.** The key failure mode under a bitshift trellis is that the decoder repeatedly queries $g(\cdot)$ on inputs that share $L - \Delta$ bits. If the generator leaves the shared bits weakly mixed into the low-order bytes, then many correlated pairs $(x_t, x_{t+1})$ produce repeated low-byte patterns, and the joint cloud $\{(y_t, y_{t+1})\}$ concentrates on structured slices. MUL-BAL explicitly targets this by mixing *before* Gaussianization: the modular multiply in Eq. (2) spreads information across bit positions via carry interactions, and the XOR-with-shift step in Eq. (3) feeds higher-order information back into lower bits. This is essential because the Gaussianizer in Eq. (4) depends directly on the *four low-order bytes*. Intuitively, we enforce a "high-to-low feedback" path so that shared-bit overlap does not translate into deterministic structure in the low bytes, and thus does not appear as banding/arms in the joint neighbor geometry.

**Baselines and what "lookup-free" means here.** We compare against QTIP-style computed codes (1MAD and 3INST) and a hybrid computed–lookup baseline (HYB). 1MAD generates a pseudorandom 32-bit word via an LCG and then byte-sums to approximate a Gaussian; 3INST

*Table 1.* Compute-code diagnostics against a Gaussian reference (lower is better). Avg. Rank averages five per-metric ranks over deployable codes only; Gaussian is an unranked reference. Fig. 3 provides the primary diagnostic.

| Code | $|\rho_{\text{edge}}|$ | $|\rho_1|$ | $W_1$ | AD | $|\kappa - 3|$ | Avg. Rank |
|---|---|---|---|---|---|---|
| Gaussian | 0.0014 | 0.0007 | 0.0037 | 1.1 | 0.0215 | — |
| 1MAD | 0.0135 | 0.0106 | 0.0206 | 36.4 | 0.2980 | 3.2 |
| 3INST | 0.0002 | 0.0012 | 0.1917 | 3030.8 | 0.1182 | 2.4 |
| HYB | 0.0013 | 0.0022 | 0.0412 | 220.6 | 0.0511 | 2.2 |
| **MUL-BAL** | 0.0047 | 0.0052 | 0.0194 | 31.1 | 0.2839 | 2.2 |

uses a different, FP16-oriented construction. HYB uses a small cache-resident LUT (not strictly lookup-free), but is included as a strong deployment baseline to isolate whether a small LUT is sufficient to remove overlap artifacts.

**Diagnostics: separating marginal Gaussianity from overlap-induced dependence.** We evaluate a code family along three complementary axes: (i) *overlap-induced dependence* along trellis edges, (ii) *temporal neighbor dependence* under consecutive windows, and (iii) *marginal* Gaussianity. Let $\mathcal{E}$ denote the directed trellis edge set induced by fixed $(L, k, V)$, and let $y_t = g(x_t)$. We report

$$|\rho_{\text{edge}}| = |\text{Corr}(g(x), g(x'))|, \qquad (x \to x') \in \mathcal{E},$$
$$|\rho_1| = |\text{Corr}(y_t, y_{t+1})|. \tag{6}$$

To quantify marginal Gaussianity, we report $W_1(y, \mathcal{N}(0,1))$, the Anderson–Darling statistic (AD; tail-sensitive), and $|\kappa - 3|$ (excess kurtosis magnitude). All metrics are "lower is better". Table 1 aggregates these signals. Since the metrics have different scales and units, we avoid an additive score and report **Avg. Rank**, the mean per-metric rank among deployable codes, only as a coarse ordering. Because these metrics capture complementary effects and the codes trade off across them, we treat the joint reachable-set geometry in Fig. 3 as the *primary* diagnostic.

**Quantitative diagnostics.** The Gaussian reference marks the target regime: near-zero dependence and near-perfect marginal tests. Among computed, strictly lookup-free options, **MUL-BAL** provides the best overall operating point: it keeps marginal deviations small ($W_1$/AD) while substantially reducing overlap-induced dependence relative to 1MAD (notably in $|\rho_{\text{edge}}|$). Two failure modes are also explicit. First, small correlations alone are insufficient: 3INST attains low $|\rho|$ yet catastrophically fails tail-sensitive marginal tests (AD), consistent with discrete-support / axis-arm artifacts in joint space. Second, a small LUT is not a complete remedy under overlap: HYB improves dependence but still exhibits much worse marginal metrics than MUL-BAL, indicating that overlap-robust mixing and marginal Gaussianity must be enforced jointly.

**Qualitative geometry.** Figure 3 visualizes the *reachable-set geometry* $\{(y_t, y_{t+1})\}$ under identical trellis settings.

A good code should fill the plane densely with near-isotropic coverage (as the Gaussian reference does). **MUL-BAL** produces the closest match among deployable baselines, indicating effective mixing of shared bits. By comparison, 1MAD shows clear diagonal banding (overlap-amplified slices), 3INST forms pronounced cross/arm structure (discrete-support artifacts), and HYB exhibits axis-aligned grid/striping consistent with hashed indexing into a finite LUT under overlap. These structures are not merely aesthetic: they indicate poor joint coverage, reducing the effective diversity of realizable neighbor pairs under the trellis constraint.

MUL-BAL is designed and validated as an *overlap-robust* computed Gaussian code: it preserves strong 1D Gaussianity while explicitly improving the 2D neighbor geometry induced by bitshift overlap. This generator is the lookup-free foundation that Stage I can select discretely per layer and Stage III can refine via per-layer affine relaxation, without ever introducing inference-time codebook loads.

### 3.2. Stage I: Discrete structural selection and trellis index packing

**Outer loop: *fixed* 3-candidate pool (pre-vetted LCG multipliers + balance shifts).** Stage I selects the discrete code parameters $\theta_\ell = (a_\ell, s_\ell)$ for each layer from a tiny, pre-vetted pool:

$$\mathcal{C} = \left\{ (a, s) \right\}$$
$$= \left\{ (1927765585, 17), (1630827939, 17), \right.$$
$$\left. (517494369, 13) \right\}, \quad |\mathcal{C}| = 3. \tag{7}$$

The three 32-bit odd multipliers $a$ are chosen *once* offline from a large spectral-test search space following standard congruential-generator screening (i.e., strong lattice structure under the spectral test in low dimensions and lagged settings), and then further filtered by our overlap-aware diagnostics (Table 1, Fig. 3) to retain only candidates that remain robust under bitshift-window overlap. This design makes Stage I cheap and reproducible: we enumerate only three candidates per layer while still allowing layer-wise specialization.

**Curvature proxy and incoherence transform (implementation-aligned).**

Following the second-order PTQ practice, we employ a curvature proxy derived from calibration statistics for each layer. Concretely, we load a symmetric packed representation flatH and mean vector $\mu$ from calibration, reconstruct

$$H_\ell \approx \text{UNPACK}(\text{flatH}) + \mu\mu^\top, \tag{8}$$

and apply standard damping/regularization $H_\ell \leftarrow H_\ell + \lambda I$ for numerical stability. Here UNPACK simply reconstructs

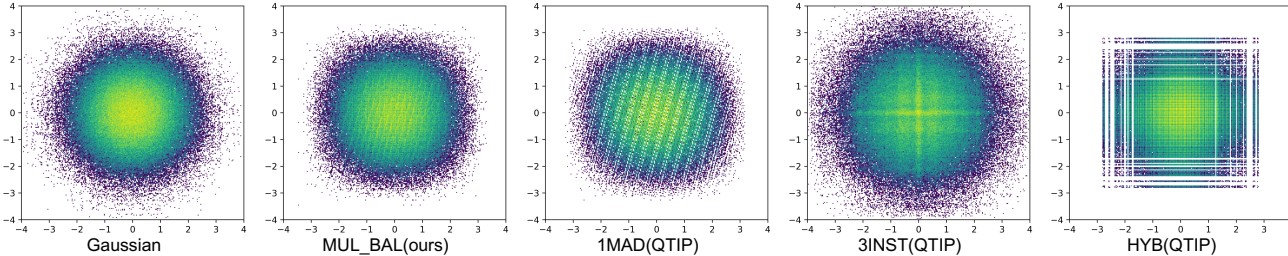

*Figure 3.* Neighbor reachable-set heatmaps $\{(y_t, y_{t+1})\}$ under identical trellis settings. Left-to-right: Gaussian reference, **MUL-BAL** (ours), and QTIP-style baselines (1MAD, 3INST, HYB). MUL-BAL yields the densest and most isotropic joint coverage among deployable options, while competing methods exhibit visible overlap-amplified structures (banding/arms/grids) that signal insufficient mixing or discrete support.

the full symmetric proxy from its compact storage: at calibration time we retain only the lower-triangular entries of the second-moment matrix as the flat vector flatH (together with the mean vector $\mu$), and UNPACK writes them back into a dense symmetric matrix, to which the rank-one mean correction $\mu\mu^\top$ is added. This is purely a memory-efficient storage convention used during calibration and plays *no* role at inference—the deployed bitshift-trellis decoder never invokes UNPACK. We then apply a randomized Hadamard incoherence transform using Rademacher sign vectors on the input/output dimensions and fast Walsh–Hadamard transforms (FWHT), obtaining $W_{r,\ell}$ and the matched-basis curvature $H_{r,\ell}$. This ensures the Stage I proxy and solver operate *in the incoherent basis* used by Hadamard-style PTQ pipelines.

**Inner loop: Hessian-weighted TCQ under bitshift overlap.** Given $\theta$, we solve for trellis indices/bitstream $Z$ that minimize a Hessian-weighted quadratic proxy in the incoherence-processed basis:

$$\min_{\theta \in \mathcal{C}} \min_Z \ \mathcal{J}(Z, \theta) = \frac{\left\| W_r - \mathcal{D}_\theta(Z) \right\|_{H_r}^2}{\left\| W_r \right\|_{H_r}^2}, \qquad (9)$$

$$\text{where} \quad \|A\|_H^2 \triangleq \text{Tr}(A^\top H A).$$

where $W_r$ and $H_r$ denote the incoherence-processed weights and the corresponding curvature proxy in that basis. Directly optimizing Eq. (9) is intractable; we therefore use a nested search: the outer loop enumerates $\theta \in \mathcal{C}$ (3 candidates), while the inner loop solves for $Z$ using a dynamic-programming TCQ solver (Viterbi-style) integrated in a BlockLDLQ blockwise procedure.

## 3.3. Stage II: Layer-wise continuous adaptation (channel compensation)

**Motivation.** A quadratic proxy cannot fully capture the true loss in the 2-bit regime, and incoherence is not a certificate of perfect local Gaussianity. Stage II therefore adds minimal, differentiable compensation while keeping the discrete decoder unchanged.

---

**Algorithm 1** Stage I: Discrete solver and index generation

**Input:** weights $W$, curvature proxy $H$, candidates $\mathcal{C}$ over $\theta = (a, s)$, headroom $\rho$.
**Output:** packed indices/bitstream $Z^\star$, selected code $\theta^\star$.
$W_r, H_r \leftarrow \text{INCOHERE}(W, H); \quad H_r \leftarrow H_r + \lambda I.$
$\sigma_W \leftarrow \text{std}(W_r); \quad \mathcal{J}^\star \leftarrow \infty.$
**for** each $\theta \in \mathcal{C}$ **do**
    $\sigma_g \leftarrow \text{std}(g_\theta(x))$ over $x \in \{0, \ldots, 2^L - 1\}$.
    $\alpha \leftarrow \rho\,\sigma_W / \sigma_g$         (offline scale match)
    $(Z_\theta, \widehat{W}_\theta) \leftarrow \text{BLOCKLDLQ-TCQ}(W_r, H_r, g_\theta, \alpha).$
    $\mathcal{J}_\theta \leftarrow \|W_r - \widehat{W}_\theta\|_{H_r}^2 / \|W_r\|_{H_r}^2.$
    **if** $\mathcal{J}_\theta < \mathcal{J}^\star$ **then**
        $\mathcal{J}^\star \leftarrow \mathcal{J}_\theta; \quad \theta^\star \leftarrow \theta; \quad Z^\star \leftarrow Z_\theta.$
    **end if**
**end for**
**Output:** freeze $\{Z^\star, \theta^\star\}$ for deployment.

---

**Learnable channel scales (QuIP#/QTIP-style).** We freeze $\{Z_\ell^\star, \theta_\ell^\star\}$ and learn only two diagonal per-channel scales $S_{\text{in}}$ and $S_{\text{out}}$ (corresponding to QuIP#/QTIP's $S_U, S_V$):

$$\tilde{W} = \text{diag}(S_{\text{out}}) \cdot \mathcal{D}_{\theta^\star}(Z^\star) \cdot \text{diag}(S_{\text{in}}),$$

$$\min_{S_{\text{in}}, S_{\text{out}}} \ \mathbb{E}_{X \sim \mathcal{D}_{\text{calib}}} \left\| f_\ell(X; W_\ell) - f_\ell(X; \tilde{W}_\ell) \right\|_F^2. \qquad (10)$$

At deployment, $S_{\text{in/out}}$ are per-layer constants and do not modify the fixed trellis decoder. When the graph permits, these diagonal scales can be folded into neighboring normalization gains or adjacent linear weights.

## 3.4. Stage III: Global few-shot relaxation (distillation)

**Frozen discrete structure; learnable continuous statistics.** Stage III keeps the discrete structure *fully frozen*: trellis indices/bitstream $Z_\ell$ and discrete generator parameters $\theta_\ell = (a_\ell, s_\ell)$ remain unchanged to preserve hardware compatibility. We only unlock a tiny set of continuous statistics *per layer*: the MUL-BAL affine parameters $(\mu_\ell, \sigma_\ell)$ inside the Gaussianizer and (optionally) the channel scales

**Algorithm 2** Stage II: Continuous adaptation (channel compensation)

---

**Input:** frozen $\{Z^\star, \theta^\star\}$; calibration data $\mathcal{X}$.
**Output:** channel scales $S_{\text{in}}, S_{\text{out}}$.
**for** each layer $\ell$ **do**
    Initialize $S_{\text{in}}, S_{\text{out}}$; freeze $Z^\star, \theta^\star$.
    **for** batch $B \in \mathcal{X}$ **do**
        $Y_{\text{ref}} \leftarrow f_\ell(B; W_{\text{orig}})$.
        $\tilde{Y} \leftarrow f_\ell\Big(B; \tilde{W}(Z^\star, \theta^\star, S_{\text{in}}, S_{\text{out}})\Big)$.
        $\mathcal{L} \leftarrow \|Y_{\text{ref}} - \tilde{Y}\|_F^2$; update $S_{\text{in}}, S_{\text{out}}$ (SGD/Adam).
    **end for**
**end for**

---

**Algorithm 3** Stage III: Global few-shot relaxation (layer-wise affines)

---

**Input:** student $f_Q$, teacher $f_T$, calibration set $\mathcal{D}_{calib}$, epochs $K$.
**Output:** per-layer affines $\{(\mu_\ell, \sigma_\ell)\}_\ell$ (and optional channel scales).
Freeze discrete: $\{Z_\ell, \theta_\ell = (a_\ell, s_\ell)\}_\ell$ and trellis $(L, k, V)$.
Unlock continuous: $\{(\mu_\ell, \sigma_\ell)\}_\ell$ (and optional Stage II scales).
**for** epoch $= 1$ to $K$ **do**
    **for** batch $x \in \mathcal{D}_{calib}$ **do**
        $\widehat{W} \leftarrow \text{DECODE}(\{Z_\ell, \theta_\ell, \mu_\ell, \sigma_\ell\}_\ell, \text{scales})$.
        $Y_s \leftarrow f_Q(x; \widehat{W})$;   $Y_t \leftarrow f_T(x)$.
        $\mathcal{L} \leftarrow \text{KL}(Y_t \| Y_s)$; update continuous params only (AdamW).
    **end for**
**end for**
Fuse each layer's $(\mu_\ell, \sigma_\ell)$ into inference-time kernel constants.

---

from Stage II. Importantly, *each layer has its own* $(\mu_\ell, \sigma_\ell)$; we do **not** share a single global affine pair across the entire model.

**Objective.** Given a small unlabeled calibration set $\mathcal{D}_{calib}$, we distill the FP16 teacher $f_T$ into the quantized student $f_Q$ by minimizing KL divergence.

**Summary and deployment invariants.** Overall, PROTEUS preserves strict deployment invariants—fixed bitshift-trellis decoding, frozen packed indices $Z_\ell$, and zero runtime LUT loads—while allowing only a minimal set of *per-layer* constants to adapt. In §4, we show that this lookup-free, overlap-robust design retains TCQ's shaping benefits and improves both accuracy and end-to-end decoding efficiency under the same fixed kernel and trellis settings.

# 4. Experiments

We evaluate PROTEUS along three axes: (i) language-model perplexity (PPL) as a rate–distortion proxy, (ii) zero-shot downstream accuracy on standard LM benchmarks, and (iii) end-to-end decoding throughput and observed memory bandwidth. Our goal is to validate that a *strictly lookup-free* TCQ realization under the bitshift-trellis constraint can improve both *accuracy* and *deployment efficiency* without changing the fixed decoder structure at inference time.

## 4.1. Experimental Setup

**Models and quantization.** We report results on Llama 2 models (7B/13B/70B) under *weight-only* PTQ: activations remain in FP16/bfloat16 while linear weights are quantized. Our main comparisons focus on 2-bit weights, where deployment constraints and solver locality are most severe.

**Benchmarks and metrics. Perplexity (PPL).** We measure PPL on WikiText2 and C4 with context length 4096 for Llama 2 (Merity et al., 2017; Raffel et al., 2020; Touvron et al., 2023). **Zero-shot accuracy.** We evaluate ARC-Challenge (ArcC), ARC-Easy (ArcE), PIQA, and Wino-Grande (WINO) using a standard LM evaluation protocol (Clark et al., 2018; Bisk et al., 2020; Sakaguchi et al., 2020; Biderman et al., 2024). **Systems metrics.** We report decoding throughput (Tok/s) and the observed effective memory bandwidth (GB/s) during decoding, to characterize the memory-traffic footprint of dequantization kernels in the latency-sensitive regime (Kwon et al., 2023).

**Baselines.** We compare against representative ultra-low-bit PTQ methods spanning (i) second-order, structured rounding families (OmniQ, QuIP, AQLM, QuIP#) and (ii) trellis-coded quantization with bitshift-trellis decoding (QTIP), plus an FP16 reference. Unless otherwise specified, we use QTIP's default HYB variant. For PPL, we report the strongest and most protocol-aligned 2-bit baselines (QuIP# and QTIP) under our evaluation pipeline; for zero-shot accuracy we include a broader set of widely-used 2-bit methods to contextualize downstream behavior.

## 4.2. Main Results

### 4.2.1. LANGUAGE MODELING PERPLEXITY

Table 2 reports PPL on WikiText2 and C4 across model sizes. At 2-bit weight-only PTQ, PROTEUS consistently improves over QuIP# and QTIP on both corpora. Beyond being a pure "systems" improvement, these gains are consistent with our diagnostics in Fig. 1 and Fig. 3: local tiles can deviate from an i.i.d. Gaussian reference after incoherence, and overlap-amplified anisotropy in the compute generator can reduce effective trellis coverage. By enforcing overlap-robust mixing (MUL-BAL) and layer-wise instan-

*Table 2.* Perplexity (↓) on WikiText2 and C4 for Llama 2.

| Method | Bits | Wikitext2 | | | C4 | | |
|--------|------|------|------|------|------|------|------|
| | | 2-7 | 2-13 | 2-70 | 2-7 | 2-13 | 2-70 |
| FP16 | 16 | 5.12 | 4.57 | 3.12 | 6.63 | 6.05 | 4.97 |
| QuIP# | 2 | 6.19 | 5.35 | 3.91 | 8.16 | 7.20 | 5.71 |
| QTIP | 2 | 5.91 | 5.26 | 3.78 | 7.76 | 6.99 | 5.56 |
| **Proteus** | 2 | **5.85** | **5.10** | **3.69** | **7.68** | **6.82** | **5.46** |

tiation, PROTEUS improves the *local* geometry the TCQ solver actually encounters.

### 4.2.2. ZERO-SHOT ACCURACY

Table 3 reports zero-shot accuracy. We use the default task definitions, prompting templates, and aggregation rules from the LM Evaluation Harness for all tasks. PROTEUS matches or improves upon QTIP/QuIP#-style 2-bit baselines across most tasks and model sizes, with consistent gains on ARC variants at 7B/13B and strong PIQA results. These results align with the geometric story isolated by our visual diagnostics: marginal Gaussianity alone does not guarantee good overlap behavior, whereas improving the neighbor reachable-set coverage (Fig. 3) reduces structured quantization texture that otherwise propagates through downstream evaluation.

### 4.2.3. ABLATION STUDIES

To isolate where the gains come from, Table 4 ablates (i) the *code realization* (lookup HYB vs. compute-only), and (ii) the incremental benefit of Stage I–III in PROTEUS on Llama2-7B W2 (WikiText2 PPL). Swapping QTIP-HYB (lookup) to our compute-only MUL-BAL already reduces PPL substantially, indicating that overlap-robust mixing is a first-order driver. Stage I+II yields an additional improvement, consistent with the need for *layer-aware* instantiation plus lightweight channel compensation under local mismatch. Stage III provides a smaller but consistent final gain via global few-shot relaxation without changing the deployed discrete structure. Finally, applying Stages I–III on top of QTIP-1MAD improves over QTIP-HYB but still trails MUL-BAL, reinforcing that *the generator geometry under overlap* matters beyond marginal Gaussianity.

### 4.2.4. DECODING THROUGHPUT AND MEMORY TRAFFIC

All throughput experiments are conducted on a *single* NVIDIA H800 (3.4 TB/s peak HBM bandwidth) with batch size 1. We report steady-state *decode* throughput (token generation) after warmup, excluding prompt-prefill, and compute memory bandwidth over the same decode window. Table 5 reports decoding throughput and the *observed* effec-

tive memory bandwidth during decoding. Across 2–4 bits, PROTEUS matches or slightly improves Tok/s relative to QTIP, while *reducing* memory BW. Notably, the observed bandwidth spans roughly 0.57–2.04 TB/s (about 17–60% of peak) at batch size 1, indicating that dequantization remains strongly memory-traffic sensitive in the single-stream regime. This behavior is consistent with our design goal: eliminating inference-time codebook loads avoids irregular on-chip memory access in the dequantization inner loop, yielding a more cache- and bandwidth-efficient decode path without sacrificing end-to-end throughput.

**Batch scaling and the crossover at** $BS \geq 32$**.** Table 5 reports the single-stream ($BS = 1$) regime, where a strong lookup baseline's codebook is cache-resident and both kernels are HBM-bound, so PROTEUS and QTIP achieve comparable throughput. The lookup-free advantage is a *batch-amortization* effect: each decoded weight is reused across $BS/8$ matmul passes per K-tile, so as the batch grows QTIP's fixed shared-memory codebook traffic becomes the dominant cost while PROTEUS's arithmetic decode does not. Table 6 shows the resulting trajectory: the two methods are comparable at $BS = 1$, PROTEUS is slightly behind at $BS = 16$ (the arithmetic decode is not yet fully amortized), then overtakes QTIP at $BS = 32$ (2940 vs. 2797 tok/s, $1.05\times$) and remains on par at $BS = 64$. A microarchitectural account of *why* this crossover occurs—shared-memory port saturation in the QTIP decode path—is given in Appendix C.

**Fairness of the large-batch baseline.** QTIP's public release ships only a $BS = 1$ decode kernel. For the large-batch comparison we implemented a batch-extended QTIP kernel that preserves its exact codebook decode path and receives the *same* kernel-level optimizations as PROTEUS (512 threads, 128 registers/thread, full-buffer reduction, decode-once-reuse). This improved QTIP's $BS = 64$ per-layer latency by $4.2\times$ over a naive configuration; Table 6 compares against this strongest QTIP baseline.

## 5. Conclusion

We presented PROTEUS, a strictly lookup-free trellis-coded quantization (TCQ) framework for ultra-low-bit LLM PTQ under hardware-friendly bitshift-trellis decoding. By mitigating overlap-induced dependence with the **MUL-BAL** compute generator, PROTEUS preserves deployment invariants (fixed decoder and packed indices) while improving 2-bit quality across the Llama 2 family. Our results emphasize that overlap-robust *joint* reachable-set geometry—not marginal distribution matching alone—is central to accurate, bandwidth-efficient TCQ deployment on modern GPUs and emerging accelerators.

*Table 3.* Zero-shot accuracy (%, higher is better) for Llama 2 models.

| | 2-70 | | | | | 2-13 | | | | | 2-7 | | | | |
|---|---|---|---|---|---|---|---|---|---|---|---|---|---|---|---|
| Method | Bits | ArcC | ArcE | PIQA | WINO | Bits | ArcC | ArcE | PIQA | WINO | Bits | ArcC | ArcE | PIQA | WINO |
| FP16 | 16 | 51.1 | 77.7 | 81.1 | 77.0 | 16 | 45.6 | 73.3 | 73.5 | 69.6 | 16 | 40.0 | 69.3 | 78.5 | 67.3 |
| OmniQ | 2 | 28.7 | 55.4 | 68.8 | 53.2 | 2 | 23.0 | 44.4 | 62.6 | 52.6 | 2 | 21.6 | 35.2 | 57.5 | 51.5 |
| QuIP | 2 | 34.0 | 62.2 | 74.8 | 67.5 | 2 | 23.5 | 45.2 | 62.0 | 52.8 | 2 | 19.4 | 26.0 | 54.6 | 51.8 |
| AQLM | 2 | 47.9 | 77.7 | 80.4 | 75.9 | 2 | 38.5 | 67.0 | 75.1 | 69.5 | 2 | 33.6 | 62.8 | 73.5 | 64.6 |
| QuIP# | 2 | 48.7 | 77.3 | 80.3 | 75.9 | 2 | 39.5 | 69.3 | 77.3 | 67.7 | 2 | 34.6 | 64.6 | 75.1 | 64.9 |
| QTIP | 2 | 48.1 | 76.9 | 80.1 | 76.5 | 2 | 39.2 | 70.6 | 77.8 | 71.0 | 2 | 35.3 | 63.9 | 75.3 | 66.7 |
| **Proteus** | 2 | **49.0** | **77.0** | **81.2** | **76.8** | 2 | **40.0** | **71.7** | 77.8 | 71.0 | 2 | **36.1** | 64.6 | **75.9** | **67.2** |

*Table 4.* Ablation on Llama2-7B W2 (WikiText2 PPL, lower is better).

| Variant | PPL ↓ |
|---|---|
| QTIP-HYB (lookup / micro-LUT) | 5.9100 |
| MUL-BAL (ours, compute-only) | 5.8792 |
| MUL-BAL + Stage I + Stage II | 5.8594 |
| MUL-BAL + Stage I + Stage II + Stage III | **5.8521** |
| QTIP-1MAD + Stage I + Stage II + Stage III | 5.8794 |

*Table 5.* Decoding throughput (Tok/s, higher is better) and observed memory bandwidth (GB/s, lower is better) for end-to-end decoding.

| | | 2–7B | | 2–70B | |
|---|---|---|---|---|---|
| Method | Bits | Tok/s | BW (GB/s) | Tok/s | BW (GB/s) |
| **Baselines** | | | | | |
| FP16 | 16 | 125.04 | — | OOM | — |
| **QTIP** | | | | | |
| QTIP | 2 | 159.13 | 710 | 36.29 | 1020 |
| QTIP | 3 | 155.33 | 950 | 33.59 | 1350 |
| QTIP | 4 | 157.46 | 1340 | 35.33 | 2040 |
| **Proteus (ours)** | | | | | |
| Proteus | 2 | 161.07 | 572 | 36.28 | 740 |
| Proteus | 3 | 155.24 | 770 | 33.84 | 1010 |
| Proteus | 4 | 157.13 | 1120 | 35.45 | 1490 |

## 6. Limitations

We highlight three limitations that scope the claims of this work.

**Scope of generality.** Our three-candidate generator pool was screened once offline and transfers across LLaMA-2/3 and Qwen2.5/3. We have not tested structurally different regimes, such as MoE, very-long-context models, or non-Transformer backbones. MUL-BAL assumes randomized Hadamard preprocessing makes weights approximately i.i.d. Gaussian; otherwise its generator distribution may mismatch. Thus, gains should be read within incoherence-based PTQ.

**Untested ALU-constrained hardware.** MUL-BAL replaces table lookups with lightweight integer arithmetic, which overlaps with tensor-core matmul via INT32 on our GPUs. On accelerators with limited or shared integer ALUs, decode arithmetic may matter and shift the compute–memory trade-off. We have not measured PROTEUS there.

**Batch-size regime of efficiency gains.** PROTEUS's accuracy gains are batch-size independent, but its efficiency advantage over lookup decoding mainly emerges at BS≥ 32 (Table 6). At BS= 1, codebooks are cache-resident and both kernels are largely HBM-bound, so throughput is comparable.

## Impact Statement

This paper presents work whose goal is to advance the field of Machine Learning. There are many potential societal

*Table 6.* End-to-end decode throughput for Llama-2-7B W2 across batch sizes. Speedup is PROTEUS/QTIP.

| BS | QTIP | Proteus | Speedup |
|---|---|---|---|
| 1 | 159 | 161 | 1.01× |
| 16 | 2126 | 2011 | 0.95× |
| 32 | 2797 | **2940** | **1.05×** |
| 64 | 3605 | 3602 | 1.00× |

consequences of our work, none which we feel must be specifically highlighted here.

## Acknowledgements

We thank the anonymous reviewers and the area chair for their insightful comments and constructive suggestions. We also gratefully acknowledge the support from the PaddlePaddle team and the National Engineering Research Center of Deep Learning Technology and Application.

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

## A. Generalization to Newer Architectures

To further test whether the selected compute-code pool is specific to Llama 2, we evaluate the same unmodified three-candidate pool on newer decoder-only architectures. Table 7 reports 2-bit weight-only MMLU accuracy on Qwen2.5-7B-Instruct and Llama-3-8B-Instruct. Across both models, PROTEUS improves over QTIP under the same candidate pool, suggesting that the pool is not overfit to the Llama 2 family and that the overlap-robust MUL-BAL design transfers to more recent model architectures.

*Table 7.* Generalization to newer architectures: 2-bit weight-only MMLU (%, 5-shot, higher is better) using the *same* unmodified 3-candidate pool. $\Delta$ is measured in MMLU points.

| Model | Bits | QTIP | Proteus | $\Delta$ |
|---|---|---|---|---|
| Qwen2.5-7B-Instruct | 2 | 59.71 | **60.98** | +1.27 |
| Llama-3-8B-Instruct | 2 | 53.29 | **54.46** | +1.17 |

## B. Comparison with Lattice- and Entropy-Coded Quantizers

Following the suggestion raised during discussion, we evaluated a nested-lattice quantizer (NestQuant (Savkin et al., 2025)) and an entropy-coded compressor (EntroLLM (Sanyal et al., 2025)) under our 2-bit weight-only protocol. As summarized in Table 8, neither method remains stable at 2-bit weight-only: perplexities exceed $10^3$ across all model sizes. We attribute this to a regime mismatch rather than an implementation artifact. NestQuant is designed as a *joint* weight/activation/KV quantizer on the Gosset ($E_8$) lattice and reports its strongest results at 4 bits; EntroLLM is a *lossless* entropy-coding layer applied on top of a 4–8-bit base quantizer for edge deployment, so at 2 bits its quality is bounded by a base quantizer operating far outside its intended range. For EntroLLM the lowest bit-width at which we obtain usable models is 4-bit (shown for context), which is not directly comparable to the 2-bit setting that PROTEUS targets. These results motivate our choice of QuIP# and QTIP as the primary 2-bit weight-only baselines in the main text.

## C. Microarchitectural Profiling

We profile both decode kernels with Nsight Compute (shape $4096 \times 11008$, Llama-2-7B 2-bit) on the same GPU used for Table 5, at $BS \in \{1, 16, 32, 64\}$. At $BS = 1$ both kernels are HBM-bound and behave almost identically (SM throughput 73.6 vs. 73.7%). As the batch grows, QTIP's 512-entry codebook—replicated $32\times$ in shared memory (64 KB) to avoid bank conflicts—requires four shared-memory lookups per decoded weight, and this traffic scales with the number of matmul passes. The result is shared-memory port sat-

*Table 8.* Lattice- and entropy-coded baselines under our 2-bit weight-only pipeline (WikiText2 / C4 PPL, lower is better). At 2-bit both methods are outside their design regime and fail to converge to a usable model. EntroLLM's 4-bit results, where it produces usable models, are listed for context only and are not directly comparable to the 2-bit setting.

| Method | Bits | WikiText2 | | | C4 | | |
|---|---|---|---|---|---|---|---|
| | | 2-7 | 2-13 | 2-70 | 2-7 | 2-13 | 2-70 |
| NestQuant | 2 | *diverged* ($> 10^2$ on all sizes) | | | | | |
| EntroLLM | 2 | *diverged* ($> 10^2$ on all sizes) | | | | | |
| EntroLLM | 4 | 8.72 | 5.49 | 4.19 | 10.69 | 7.16 | 6.65 |
| **Proteus** | 2 | **5.85** | **5.10** | **3.69** | **7.68** | **6.82** | **5.46** |

*Table 9.* Nsight Compute warp-stall and SM-throughput profiling (SM90a). Lower stall percentages are better; higher SM throughput is better.

| Metric | Method | BS=1 | BS=16 | BS=32 | BS=64 |
|---|---|---|---|---|---|
| MIO Throttle (%) | QTIP | 1.2 | 2.3 | 12.6 | 30.5 |
| | **Proteus** | 5.7 | 3.6 | 3.6 | 2.4 |
| Short Scoreboard (%) | QTIP | 0.9 | 2.6 | 6.1 | 19.5 |
| | **Proteus** | 1.9 | 2.3 | 4.1 | 2.9 |
| SM Throughput (%) | QTIP | 73.7 | 45.0 | 22.1 | 17.8 |
| | **Proteus** | 73.6 | 67.8 | 54.4 | 32.9 |

uration: QTIP's MIO Throttle rises to 30.5% at $BS = 64$ ($12.5\times$ PROTEUS), and its Short Scoreboard stalls (shared-memory$\rightarrow$register dependencies feeding the tensor cores) reach 19.5% ($6.8\times$ PROTEUS); together these account for roughly half of QTIP's warp cycles at $BS = 64$. PROTEUS's arithmetic decode issues about $2.2\times$ more INT32 instructions but uses *zero* shared memory for the codebook; on SM90a the INT32 pipe is physically separate from both the tensor-core and MIO pipes, so decode overlaps matmul and MIO Throttle stays flat at 2–4%. Both kernels issue identical tensor-core MMA counts and both saturate DRAM throughput at 100% across all batch sizes, confirming that the gap arises entirely from the decode path rather than from the matmul or DRAM utilization. Consistent with efficient compute, PROTEUS's dominant stall at $BS = 64$ shifts to Long Scoreboard (61.2% vs. QTIP's 23.4%)—i.e., cleanly waiting on the next HBM batch rather than on on-chip contention.

