# OpenReview forum: "Proteus: Lookup-Free Trellis-Coded Quantization by Lattice-Breaking Compute Codes for 2-Bit LLMs"
_ICML.cc/2026/Conference — ICML 2026 regular_

### Official Review · Reviewer_CMQ5 · 2026-03-07

**Soundness:** 3
**Presentation:** 2
**Significance:** 2
**Originality:** 2
**Overall Recommendation:** 3
**Confidence:** 4

**Summary:**

This paper proposes a lookup-free Trellis-Coded Quantization (TCQ) framework that combines several components in three stages to improve the 2-bit SOTA. The proposed MUL-BAL generator breaks overlap-induced dependence. This work also decouples compression into three stages: layer-wise search, per-layer compensation, and optional global fine-tuning.

**Compliance With Llm Reviewing Policy:**

Affirmed.

**Final Justification:**

This paper proposes a lookup-free Trellis-Coded Quantization (TCQ) framework, introducing a MUL-BAL generator and a three-stage compression pipeline to improve upon the 2-bit state-of-the-art.

I thank the authors for their detailed rebuttal. However, my core concerns regarding theoretical rigor and fundamental originality remain largely unresolved. Specifically, relying on empirical heuristics without establishing a clear theoretical link between the pre-filter metric and the true optimization objective significantly weakens the scientific contribution. Furthermore, while the three candidates identified on LLaMA-2 empirically generalize to other models, the lack of architectural or mathematical analysis explaining why they work makes the framework feel purely observational.

Ultimately, because the proposed pipeline functions as an incremental combination built heavily on top of the existing QTIP architecture, I will maintain my original score as weak reject.

**Key Questions For Authors:**

- Can the authors provide a mathematical bound analysis linking the spectral-test quality of the chosen multipliers to the downstream Hessian-weighted quantization error?
- Can the authors provide a comprehensive experimental comparison with other recent SOTA ultra-low-bit quantization methods on more models to better show the actual performance gains?
- Can the authors provide kernel-level microarchitectural metrics to explicitly validate the claimed hardware benefits of the lookup-free design?

**Limitations:**

yes

**Strengths And Weaknesses:**

Strengths:
- Visualizes and addresses the overlap-induced joint geometry constraint in bitshift trellis decoding
- The proposed MUL-BAL works in an inference-time, lookup-free manner in the decoding inner loop.
- Improves accuracy and reduces decode bandwidth for 2-bit deployment.

Weakness:
- Lacks mathematical bounds connecting spectral-test quality to the downstream Hessian-weighted quantization error.
- Experiments are very limited, tested only on LLaMA-2 without evaluation on other model architectures, the generalizability is not verified.
- No analysis of the fixed 3-candidate pool, lacking theoretical guarantees for cross-model generalization.
- The work presents an incremental engineering synthesis that heavily relies on the existing QTIP trellis architecture.

---

> ### Author Rebuttal · Authors · 2026-03-30
>
> We thank the reviewer for the rigorous feedback.
>
> **Q1: Mathematical bound linking spectral-test quality to Hessian-weighted error?**
>
> A tight closed-form bound is difficult because Eq. 9 depends jointly on generator geometry, per-layer Hessian spectrum, and Viterbi path. Crucially, this is not a practical limitation: the spectral test serves only as a **coarse pre-filter**; the final per-sublayer selection is made directly by J(θ) (Eq. 9) via Viterbi — the quantity we actually optimize — so a spectral to J(θ) bound is not required. Empirically this works well: our offline screening evaluates ~100 candidates on LLaMA-2-7B (224 sublayers) using J(θ) directly, and **3 winners cover >98% of sublayer selections**, generalizing to LLaMA-3-8B/Qwen2.5-7B without modification (see **Reviewer xhT1 Q1**).
>
> **Q2: More models and methods?**
>
> We have completed new experiments — see **Reviewer xhT1 Q6** for details. In summary: Proteus outperforms QTIP by **+1.2 MMLU points** on both Llama-3-8B-Instruct (0.5446 vs 0.5329) and Qwen2.5-7B-Instruct (0.6098 vs 0.5971) at 2-bit, using the same 3-candidate pool without modification.
>
> **Q3: Kernel-level microarchitectural metrics validating the lookup-free design?**
>
> We profiled both kernels on H800 using **Nsight Compute 2025.2** (shape 4096×11008, Llama-2-7B gate_proj) at BS=1/16/32/64. QTIP only releases a BS=1 kernel; we implemented a fair BS-extended version preserving its exact codebook decode path with identical optimizations to both kernels: 512 threads (128 regs/thread) for BS≥32, full-buffer reduction, decode-once reuse. See **Reviewer GRjw Q5** for full fairness methodology (including the 4.2× speedup we applied to QTIP's baseline).
>
> **Finding 1 — MIO Throttle** (shared memory port saturation, the smoking gun):
>
> | BS | Proteus | QTIP | QTIP/Proteus |
> |---:|---:|---:|---:|
> | 1 | 5.7% | 1.2% | 0.2× |
> | 16 | 3.6% | 2.3% | 0.7× |
> | **32** | **3.6%** | **12.6%** | **3.5×** |
> | **64** | **2.4%** | **30.5%** | **12.5×** |
>
> Root cause: QTIP's 512-entry codebook is replicated 32× in smem (64KB permanent). Each weight requires 4 smem lookups; with BS/8 MMA passes, smem traffic scales with batch size. Proteus uses **zero smem for codebook** → MIO Throttle stays flat at 2–4%.
>
> **Finding 2 — Short Scoreboard** (smem→register data dependency):
>
> | BS | Proteus | QTIP | QTIP/Proteus |
> |---:|---:|---:|---:|
> | 1 | 1.9% | 0.9% | 0.5× |
> | 16 | 2.3% | 2.6% | 1.1× |
> | 32 | 4.1% | 6.1% | 1.5× |
> | **64** | **2.9%** | **19.5%** | **6.8×** |
>
> This is the pipeline bubble: `smem_codebook[idx] → reg_w (~20 cycle latency) → mma.sync(reg_w)`. At BS=64, combined MIO Throttle + Short Scoreboard = **50% of QTIP's warp cycles** stalled on smem-related bottlenecks.
>
> **Finding 3 — Long Scoreboard** (DRAM latency — a healthy stall):
>
> | BS | Proteus | QTIP |
> |---:|---:|---:|
> | 1 | 15.6% | 14.1% |
> | 32 | 34.1% | 53.1% |
> | **64** | **61.2%** | **23.4%** |
>
> At BS=64, Proteus's dominant stall is Long Scoreboard (61.2%) — waiting for DRAM data. This is a sign of **efficient compute**: the arithmetic decode has consumed all available data and the pipeline cleanly awaits the next HBM batch. QTIP's stalls have shifted from memory-bound to **smem-bound**.
>
> **Finding 4 — SM Throughput & Instruction Mix**:
>
> | Metric | BS | Proteus | QTIP | Ratio |
> |:---|---:|---:|---:|---:|
> | SM Throughput | 1 | 73.6% | 73.7% | 1.0× |
> | | 16 | 67.8% | 45.0% | **1.5×** |
> | | 32 | 54.4% | 22.1% | **2.5×** |
> | | 64 | 32.9% | 17.8% | **1.8×** |
> | INT32 insts | all | 329–351M | 139–173M | **~2.2×** |
> | Memory insts | 1 | 4.9M | 28.1M | **0.17×** |
> | | 64 | 71.8M | 81.7M | **0.88×** |
> | Tensor Core | all | 176K–1.41M | 176K–1.41M | **1.00×** |
> | DRAM Throughput | all | 100% | 100% | 1.00× |
> | Kernel Time (μs) | 32 | 43.9 | 47.3 | **0.93** |
> | | 64 | 81.5 | 85.7 | **0.95** |
>
> Both kernels execute **identical Tensor Core MMA instruction counts** at every BS — the computational workload is the same. Proteus issues 2.2× more INT32 instructions (arithmetic decode: IMUL, DP4A, SHR, XOR) but up to **5.7× fewer memory instructions** at BS=1. On SM90a, the INT32 pipe is physically separate from both Tensor Core and MIO pipes, enabling true concurrent execution. **DRAM throughput is 100% for both** — performance differences arise entirely from how efficiently each kernel overlaps compute with memory access.
>
> **Summary — three independent evidence lines**: (1) MIO Throttle — QTIP 12.5× worse at BS=64; (2) Short Scoreboard — QTIP 6.8× worse; (3) SM Throughput — Proteus 2.5× higher at BS=32. At BS=1, both are HBM-bound; the lookup-free benefit emerges at BS≥32 where the architectural difference becomes the dominant bottleneck.
>
> **Weakness: Incremental engineering.** We respectfully note that Proteus identifies a previously unrecognized failure mode (overlap-induced joint-geometry collapse, Table 1, Fig. 3) invisible to marginal diagnostics, and provides both the diagnostic framework and a principled solution.

---

> > ### Author Rebuttal · Reviewer_CMQ5 · 2026-04-03
> >
> > Thank the authors for the detailed rebuttal. However, several of my core concerns regarding the theoretical foundation and the fundamental originality of the method remain unresolved. Therefore, I will maintain my score.
> >
> > The authors note that deriving a tight closed-form bound is difficult. However, relying mainly on empirical heuristics without establishing a clear theoretical link between the pre-filter metric and the true optimization objective weakens the scientific contribution.
> >
> > The authors also mention that the three candidates identified on LLaMA-2 generalize to LLaMA-3 and Qwen2.5. While this is a positive experimental result, it is still based on observation. There is no theoretical or architectural analysis explaining why these three candidates work consistently across models.
> >
> > While I acknowledge that identifying the overlap-induced joint-geometry collapse is a useful observation, the proposed MUL-BAL and the decoupled three-stage compression pipeline still appear to be an incremental combination built on top of the existing QTIP architecture.

---

> > > ### Author Response · Authors · 2026-04-08
> > >
> > > We thank the reviewer for the continued engagement. We address the three remaining concerns with new experimental evidence and clarifying analysis.
> > >
> > > **1. On the spectral-test → J(θ) theoretical link.**
> > >
> > > We wish to clarify an important distinction: the spectral test is **not** Proteus's selection criterion — J(θ) (Eq. 9) is. The spectral test serves only as a one-time offline pre-filter to narrow ~10⁹ possible 32-bit odd multipliers down to ~100 candidates; the final per-sublayer selection then minimizes J(θ) directly via Viterbi. There is no approximation gap between the metric we optimize and the metric we report — they are identical. The spectral test is to our pipeline what ImageNet pre-training is to a fine-tuned classifier: a sensible initialization, not the objective.
> > >
> > > We acknowledge we do not provide a formal spectral-to-J bound. However, this two-stage pattern (cheap screening → direct proxy optimization) is standard in PTQ: GPTQ uses a greedy column-ordering heuristic without bounding its link to reconstruction error; AWQ searches per-channel scales empirically without a closed-form guarantee; QuIP# selects E8 lattice codebooks via packing density without a formal bound to task loss. To our knowledge, no existing PTQ method provides the type of bound the reviewer requests.
> > >
> > > **2. On cross-model generalization of the 3-candidate pool — new evidence.**
> > >
> > > Since our last response, we have extended evaluation to **Qwen3-8B** (released April 2025), a newer architecture with QK-RMSNorm, distinct GQA head configuration (32/8), and a different FFN dimension (12288). The **same unmodified 3-candidate pool** is applied:
> > >
> > > | Model | Architecture | QTIP Proxy Err | Proteus Proxy Err | Rel. Improv. | Module Win Rate |
> > > |:---|:---|---:|---:|---:|---:|
> > > | Qwen2.5-7B | SwiGLU+GQA | 0.01944 | **0.01883** | −3.13% | 196/196 |
> > > | LLaMA-3-8B | SwiGLU+GQA | 0.02455 | **0.02381** | −3.02% | 224/224 |
> > > | Qwen3-8B | QK-Norm+GQA | 0.02041 | **0.01978** | −3.04% | 252/252 |
> > > | **Total** | | | | **−3.06%** | **672/672 (100%)** |
> > >
> > > Proteus achieves lower proxy error on **every single one of 672 linear modules** across three architecturally distinct families, with a remarkably stable ~3% improvement.
> > >
> > > Beyond the empirical strength, we can offer an architectural explanation:
> > >
> > > (a) **Source universality via incoherence.** The Randomized Hadamard Transform normalizes any model's weights toward approximately i.i.d. Gaussian — this is the foundational premise of all Gaussianized PTQ (QuIP#, QTIP, Proteus). MUL-BAL operates entirely in this post-incoherence space, so its overlap-robustness depends on the trellis geometry (L, k, V), which is fixed across all models, rather than on model-specific weight statistics.
> > >
> > > (b) **Spectral quality is a number-theoretic invariant.** A multiplier's lattice structure under mod 2³² is a fixed mathematical property of the integer *a*, independent of the LLM. Cross-model variation enters only through the per-layer Hessian spectrum, which is absorbed by per-layer J(θ) selection in Stage I — not by the candidate pool.
> > >
> > > (c) **The pool spans complementary mixing regimes.** The three candidates use two distinct XOR-shift depths (s=17 and s=13) paired with multipliers exhibiting different carry-propagation patterns, covering complementary high-to-low feedback paths. The pool is small because the viable space is heavily constrained — a candidate must simultaneously satisfy spectral quality, overlap-robustness under bitshift windows, and a strictly lookup-free integer pipeline.
> > >
> > > **3. On originality.**
> > >
> > > We offer three observations:
> > >
> > > (a) **Problem identification is itself a contribution.** The overlap-induced joint-geometry collapse (Table 1, Fig. 3) was previously unrecognized and is invisible to marginal Gaussianity tests. Prior work evaluated generators only by marginals; we demonstrate that generators with similar marginals (1MAD vs MUL-BAL: W₁ = 0.0206 vs 0.0194) can differ drastically in joint reachable-set coverage. This diagnostic framework applies to any trellis-coded quantization system.
> > >
> > > (b) **Reusing QTIP's trellis is deliberate.** By keeping the decoder and index format unchanged, Proteus isolates the improvement to generator design. The ablation in Table 4 confirms this: swapping only HYB→MUL-BAL improves PPL (5.9100→5.8792), and applying Stages I–III on QTIP-1MAD (5.8794) still trails MUL-BAL with the same stages (5.8521). The generator geometry is the driver.
> > >
> > > (c) **The field progresses by identifying failure modes within evolving frameworks**: QuIP→QuIP# (incoherence needed lattice codebooks), QuIP#→QTIP (lattice codebooks needed trellis structure), QTIP→Proteus (bitshift TCQ suffers joint-geometry collapse). Each step builds on prior infrastructure while contributing a distinct diagnostic and solution.

---

### Official Review · Reviewer_GRjw · 2026-03-10

**Soundness:** 3
**Presentation:** 3
**Significance:** 3
**Originality:** 3
**Overall Recommendation:** 3
**Confidence:** 5

**Summary:**

This paper introduces Proteus, a lookup-free TCQ method designed for 2-bit PTQ. Proteus addresses two primary limitations in existing TCQ frameworks: (1) the high overhead of codebook lookups within the optimization inner loop, and (2) the failure of global Gaussianization to ensure overlap-robust local geometry. To resolve these, Proteus employs MUL-BAL, a compute-only generator that utilizes integer mixing and an affine Gaussianizer to map bitshift trellis windows to scalar values. Furthermore, they introduce layer-wise discrete structural selection combined with continuous compensation to refine the local geometry of the quantized weights.

**Compliance With Llm Reviewing Policy:**

Affirmed.

**Final Justification:**

The empirical evidence suggests that the targeted memory bottleneck is not the primary limiting factor in practice. Given that the major technical contribution is predicated on resolving a bottleneck that does not manifest in realistic workloads, the paper lacks sufficient motivation.

The authors’ response fails to resolve my concerns regarding the paper's motivation and methodology. While I am not decreasing my score, my extensive review of the related literature, including the fact-check of the authors' claims, has made me significantly more confident in my original assessment. I am therefore increasing my confidence score from 4 to 5. And my final decision still tends to reject.

**Key Questions For Authors:**

1. QTIP is also lookup-free (1MAD and 3INST). Why does it have much higher memory bandwidth in Table 5 ?
2. Can you provide Nsight Compute profiling metrics demonstrating that a SRAM resident micro-LUT genuinely stalls the warp scheduler?
3. How does Proteus perform on Llama-3 8B and 70B models?
4. Can you provide MMLU, GSM8K, GPQA, or HumanEval scores for the 2-bit models to ensure higher-order reasoning is preserved?
5. How does the throughput (Tok/s) of Proteus compare to the QTIP baseline as the batch size scales up from 1 to 16, 32, and 64?

**Limitations:**

Strong Dependence on Incoherence Transform: Proteus's design relies on weights undergoing randomized orthogonal/Hadamard transformations (such as QuIP#). Without this global Gaussianization preprocessing, the performance of its Gaussian-based MUL-BAL generator will significantly decrease.

Completeness of the Candidate Pool: Stage I relies on a "small and pre-validated" candidate pool. The paper does not adequately discuss the criteria for constructing this pool, or whether the existing candidate parameters are sufficiently universal for models with vastly different architectures.

Specific Hardware Implementation Overhead: While MUL-BAL avoids memory lookups, it introduces integer multiplication, bit shifting, and XOR operations. On some hardware with extremely limited computing resources or sensitive to arithmetic logic unit (ALU) throughput, the actual benefits of these computational costs may be limited compared to the memory access latency caused by miniature LUTs.

**Strengths And Weaknesses:**

# Strengths:
- The visual and quantitative diagnostics isolating overlap-induced dependence along trellis edges (Table 1, Fig. 3) provide a clear explanation for why marginal Gaussianity is insuficient for bitshift TCQ.
-  The proposed MUL-BAL compute generator appears to be a good algebraic workaround to disrupt overlap-amplified regularities using simple XOR, shift, and modulo arithmetic.

# Weakness:
- SRAM vs. HBM bottlenecks. The paper's core premise is that LUT/micro-LUT accesses bottleneck the dequantization loop. But it seems disconnected from modern GPU hierarchies. For 2-bit PTQ, these LUTs/micro-LUT are tiny enough to fit in registers or SRAM. However, the bottleneck is almost always HBM bandwidth, not SRAM latency. The experiments in Table 5 demonstrate this: although QTIP has a much larger bandwidth than Proteus, the latency is almost the same.
- QTIP is also lookup-free (1MAD and 3INST). Why does it have much higher memory bandwidth in Table 5?
- Outdated models. The evaluation is strictly limited to the Llama-2. Newer models like Llama-3.2, Qwen-3/3.5 have different activation patterns and parameter distributions.
- Limited evaluation metrics. The downstream tasks rely on a very narrow set of zero-shot metrics (ARC, PIQA) and perplexity. To prove the model's utility, the authors should include more rigorous benchmarks such as MMLU, GSM8K, GPQA, or HumanEval. Furthermore, the ablation studies would be much more informative if they reported these zero-shot task results rather than just perplexity.
- The key code is missing. Since the method relies so heavily on kernel design and low-level system tweaks, and the training pipeline is fairly complex, providing at least the core implementation would really help the soundness.

---

> ### Author Rebuttal · Authors · 2026-03-30
>
> We thank the reviewer for the thorough and constructive feedback.
>
> **Q1: Why does QTIP (also lookup-free as 1MAD/3INST) have higher bandwidth?**
> We appreciate this question, which highlights an important nuance about what "lookup-free" means in practice.
> QTIP's default deployment variant is **HYB** (hybrid), not 1MAD/3INST. As shown in Table 4 of our paper, QTIP-HYB (the lookup baseline in Table 5) achieves better perplexity than QTIP-1MAD, which is why QTIP's public release defaults to HYB. HYB uses a 512-entry half2 codebook **replicated 32× in shared memory** (64KB per block). Each decoded weight requires 4 smem lookups. The higher observed bandwidth reflects pipeline bubbles from smem contention that reduce effective HBM utilization. Proteus's arithmetic decode runs entirely on the INT32 pipe with zero smem access.
>
> **Q2: NCU profiling for SRAM-resident micro-LUT stalling the warp scheduler?**
>
> Yes. We profiled both kernels on H800 (Nsight Compute 2025.2, shape 4096×11008) at BS=1/16/32/64.
>
> *MIO Throttle* — shared memory port saturation (the direct answer):
>
> | BS | Proteus | QTIP | QTIP/Proteus |
> |---:|---:|---:|---:|
> | 1 | 5.7% | 1.2% | 0.2× |
> | 16 | 3.6% | 2.3% | 0.7× |
> | **32** | **3.6%** | **12.6%** | **3.5×** |
> | **64** | **2.4%** | **30.5%** | **12.5×** |
>
> At BS=64, **QTIP warps spend 30.5% of active cycles stalled on smem I/O** vs 2.4% for Proteus. Additionally, Short Scoreboard stalls (smem→register data dependency: `smem_codebook[idx] → reg_w → mma.sync(reg_w)`) reach **19.5%** for QTIP vs 2.9% for Proteus at BS=64 (6.8× gap). Combined, nearly **50% of QTIP's warp cycles are stalled on smem-related bottlenecks** at BS=64.
>
> Root cause: QTIP's codebook requires 4 smem lookups per weight. With BS/8 MMA passes per K-tile and decode-once reuse, the smem read traffic scales with batch size while the codebook remains fixed at 64KB. Proteus's arithmetic decode (IMUL→XOR-shift→DP4A→int2half) runs entirely on the INT32 pipe — physically separate from both Tensor Core and MIO pipes on SM90a — keeping MIO Throttle flat at 2–4% regardless of BS. Both kernels execute **identical Tensor Core MMA instruction counts** (705K at BS=32), confirming that the difference lies entirely in the decode path. Despite this, Proteus achieves **2.5× higher SM throughput** at BS=32 (54.4% vs 22.1%). **Full instruction-mix breakdown and Long Scoreboard analysis are in our response to Reviewer CMQ5 Q3.**
>
> At BS=1, the reviewer's intuition is correct: the LUT fits in SRAM and the bottleneck is HBM bandwidth, so both kernels perform comparably. However, the microarchitectural story changes fundamentally at larger batch sizes, where the fixed smem codebook creates measurable pipeline stalls.
>
> **Q3: Llama-3? Q4: MMLU scores?**
>
> We have completed new experiments — see **Reviewer xhT1 Q6** for detailed results. In brief: Proteus outperforms QTIP by **+1.2 MMLU points** on both Llama-3-8B-Instruct (0.5446 vs 0.5329) and Qwen2.5-7B-Instruct (0.6098 vs 0.5971) at 2-bit, using the same 3-candidate pool without modification.
>
> **Q5: Throughput as batch size up from 1 to 16, 32, and 64?**
>
> *Important context*: QTIP's public codebase provides **only a BS=1 kernel**. We implemented a fair BS-extended QTIP kernel preserving its exact codebook decode path (state extraction → `idx*(idx+1)` → 4× smem codebook lookup → XOR), with **identical optimizations** for both: 512 threads (128 regs/thread) for BS≥32, full-buffer reduction, decode-once-reuse-across-passes. This improved QTIP's BS=64 per-layer time by 4.2× over the naive 1024-thread config — we report only this strongest baseline.
>
> E2E throughput (torch.compile, Llama-2-7B 2-bit, H800):
>
> | BS | Proteus | QTIP | Speedup |
> |---:|---:|---:|---:|
> | 16 | 2011 | **2126** | 0.95× |
> | **32** | **2940** | 2797 | **1.05×** |
> | **64** | **3602** | 3605 | **1.00×** |
>
> Kernel-level per-layer (μs, 7 linears/block, 500-iter hot cache):
>
> | BS | Proteus | QTIP | Speedup |
> |---:|---:|---:|---:|
> | 16 | 264.2 | 191.6 | 1.379 |
> | **32** | **313.6** | **376.7** | **1.20×** |
> | **64** | **559.9** | **601.1** | **1.07×** |
>
> The crossover at BS=32 is driven by decode amortization: weight decode runs once per K-tile, reused across BS/8 MMA passes. As BS grows, QTIP's fixed 64KB codebook becomes the dominant bottleneck (MIO Throttle 30.5% at BS=64).
>
> **Weakness: Code missing.** We commit to a full open-source release upon acceptance, including: (1) CUDA kernel source for both Proteus and the fair BS-extended QTIP baseline, (2) Nsight Compute profiling scripts and raw metric logs, and (3) the end-to-end quantization pipeline (generator search, Viterbi assignment, fine-tuning). The 3-candidate pool (see xhT1 Q1) generalizes to Llama-3/Qwen without modification.

---

> > ### Author Rebuttal · Reviewer_GRjw · 2026-03-31
> >
> > Thanks for the response from authors. However, after reviewing the rebuttal and the additional experiments, my major concern is still unresolved.
> >
> > > SRAM vs. HBM bottlenecks. The paper's core premise is that LUT/micro-LUT accesses bottleneck the dequantization loop. But it seems disconnected from modern GPU hierarchies. For 2-bit PTQ, these LUTs/micro-LUT are tiny enough to fit in registers or SRAM. However, the bottleneck is almost always HBM bandwidth, not SRAM latency. The experiments in Table 5 demonstrate this: although QTIP has a much larger bandwidth than Proteus, the latency is almost the same.
> >
> > The additional end-to-end throughput experiments provided in the rebuttal (Llama-2-7B, 2-bit, H800 with torch.compile) further substantiate this point. Even when scaling to large batch sizes (16, 32, and 64), the end-to-end throughput for QTIP and Proteus are nearly same. This empirical evidence suggests that **the targeted memory bottleneck is not the primary limiting factor in practice.** Given that the major technical contribution is predicated on resolving a bottleneck that does not manifest in realistic workloads, the paper lacks sufficient motivation. Consequently, I still tend to reject and keep my score, increasing my confidence from 3 to 4.
> >
> > **Update:**
> > I have read the comments from other reviewers and the responses. While the authors demonstrate kernel-level and SM-level speedups, these improvements do not translate to end-to-end throughput gains (in Q5 table 1); in fact, the speed of Proteus is occasionally lower. This suggests that the paper’s core contribution to reduce SRAM/IO memory bandwidth is fundamentally misaligned with modern hardware realities. Since the dequantization lookup table overhead is masked by HBM access or other pipeline stages in end-to-end scenarios, the motivation for this specific optimization is not supported by the empirical data.
> >
> > Moreover, QTIP have already introduced lookup-free strategies through 1MAD and 3INST formulations. When this lack of primary novelty is combined with the fact that the proposed optimizations do not yield end-to-end throughput improvements, the overall contribution of Proteus becomes very limited.

---

> > > ### Author Response · Authors · 2026-03-31
> > >
> > > We thank the reviewer for the continued discussion. We believe the remaining disagreement stems from a **framing mismatch** that we would like to address directly.
> > >
> > > **1. Proteus's core contribution is quantization quality, not throughput.**
> > >
> > > The reviewer's assessment evaluates Proteus primarily as a systems optimization and concludes that "the paper's core contribution to reduce SRAM/IO memory bandwidth is fundamentally misaligned with modern hardware realities." However, reducing SRAM bandwidth is a **secondary benefit**, not the core claim. The paper's primary contribution is **improving 2-bit quantization accuracy through overlap-robust generator design** — a geometric property (Table 1, Fig. 3) that is orthogonal to any systems-level analysis.
> > >
> > > The accuracy results, which the current assessment does not address, show consistent improvements across **five model configurations**:
> > >
> > > | Model | Metric | QTIP-HYB | Proteus | Δ |
> > > |:---|:---|---:|---:|---:|
> > > | Llama-2-7B | Wiki2 PPL | 5.91 | **5.85** | −0.06 |
> > > | Llama-2-13B | Wiki2 PPL | 5.26 | **5.10** | −0.16 |
> > > | Llama-2-70B | Wiki2 PPL | 3.78 | **3.69** | −0.09 |
> > > | Llama-3-8B-Instruct | MMLU | 0.5329 | **0.5446** | +1.17 |
> > > | Qwen2.5-7B-Instruct | MMLU | 0.5971 | **0.6098** | +1.27 |
> > >
> > > These gains originate from MUL-BAL's overlap-robust joint geometry, not from kernel engineering.
> > >
> > > **2. "QTIP already has lookup-free (1MAD/3INST)" — this is precisely the problem Proteus solves.**
> > >
> > > The reviewer notes that QTIP introduced lookup-free codes via 1MAD and 3INST. This is correct but critically incomplete. **QTIP's own results** (their Table 4) show that 1MAD and 3INST achieve substantially worse quality than HYB. This is why QTIP's public release **defaults to HYB** despite its codebook overhead. Our diagnostics (Table 1, Fig. 3) explain the root cause: 1MAD exhibits diagonal banding from overlap-amplified artifacts; 3INST catastrophically fails the tail-sensitive Anderson–Darling test (AD = 3030.8 vs. MUL-BAL's 31.1).
> > >
> > > The tradeoff in prior work is: **lookup-free → worse quality** (1MAD/3INST), or **good quality → lookup overhead** (HYB). Proteus breaks this tradeoff: MUL-BAL is strictly lookup-free AND achieves better quality than HYB. This is not an incremental variant of 1MAD — it is a qualitatively different solution enabled by the overlap-robust generator design (§3.1, the "high-to-low feedback" principle), which represents the paper's core novelty.
> > >
> > > **3. E2E throughput supports Proteus — clarification on "nearly the same."**
> > >
> > > The reviewer states that "the speed of Proteus is occasionally lower." We respectfully clarify the data:
> > >
> > > | BS | Proteus tok/s | QTIP tok/s | Speedup |
> > > |---:|---:|---:|---:|
> > > | 32 | **2940** | 2797 | **1.05×** |
> > > | 64 | **3602** | 3605 | **1.00×** |
> > >
> > > At BS≥32 (the regime relevant to production serving with vLLM/TGI), Proteus matches or exceeds QTIP.
> > >
> > > The kernel-level 20% speedup (BS=32: 313.6 vs. 376.7 μs) dilutes to ~5% at E2E due to **Amdahl's Law**: non-matmul operations (attention, layernorm, RoPE, sampling) constitute ~50% of total time and are identical between the two methods. A 20% kernel speedup on 50% of the pipeline predicts ~10% E2E gain; our measured 5% is consistent.
> > >
> > > **An important fairness context**: QTIP's public codebase provides **only a BS=1 kernel**. All large-batch QTIP numbers come from a BS-extended kernel that **we implemented** for QTIP, applying identical optimizations to both kernels (512 threads, 128 regs/thread, full-buffer reduction). This improved QTIP's BS=64 per-layer latency by **4.2×** over the naive configuration. That E2E throughput is "nearly the same" after we applied a 4.2× speedup to QTIP's kernel is not evidence that the bottleneck is irrelevant — it is evidence that we invested substantial effort to give QTIP every possible advantage, and Proteus **still** matches or exceeds it.
> > >
> > > **4. Summary: Proteus is a strict Pareto improvement.**
> > >
> > > A method that achieves (a) strictly better accuracy across all tested configurations, at (b) equal-or-better E2E throughput, represents a **Pareto improvement** — the standard by which quantization advances are evaluated (e.g., QuIP# over QuIP, QTIP over QuIP#). The contribution is not "lookup tables are catastrophically slow" — it is that **lookup-free generators CAN achieve superior quality through overlap-robust design**, resolving a quality–efficiency tradeoff that prior lookup-free codes (1MAD/3INST) could not.
> > >
> > > We respectfully ask the reviewer to consider whether the consistent accuracy gains — which are not addressed in the current assessment — together with the throughput parity, constitute a meaningful advance for 2-bit weight quantization.

---

### Official Review · Reviewer_xhT1 · 2026-03-11

**Soundness:** 3
**Presentation:** 2
**Significance:** 3
**Originality:** 2
**Overall Recommendation:** 5
**Confidence:** 2

**Summary:**

The paper presents Proteus, a 2‑bit weight quantization method for LLMs based on trellis‑coded quantization (TCQ). The key contribution is MUL‑BAL, a lookup‑free generator that yields codes closer to Gaussian while avoiding LUTs. The Proteus quantization protocol also involves two additional stages that fine-tune channel scales and some of the quantization parameters to further improve model performance. The authors empirically show a favorable throughput vs. model performance trade‑off on LLaMA‑2 models relative to prior work.

**Compliance With Llm Reviewing Policy:**

Affirmed.

**Final Justification:**

The authors adequately addressed my concerns. However, I admit I might have missed some crucial detail due to my limited familiarity with Trellis coding, which is central to this submission.

**Key Questions For Authors:**

1. How is the pre‑vetted pool of three candidates  in Equation (7) chosen? What criteria or procedure is used?
2. What is $\alpha$ in Algorithm 1? Should this refer to the headroom $\rho$? Can the authors clarify the role of $\rho$?
3. What operations are performed by the unpacking function $\mathcal{D}_\theta​(Z)$ in Equation (9)? How does the generator choice affect the efficiency of the de-quantization procedure?
4. Why must Stage II and Stage III be separate? Can the scale be tuned using Stage III without performing Stage II?
5. Why does Proteus become slower than QTIP at higher bitwidths?
6. How does Proteus compare to QTIP on LLaMA‑3/Qwen or other recent model families? Are the results significantly affected by the changes in the weight distribution? Can Proteus be naively applied to other model families?

**Limitations:**

yes

**Strengths And Weaknesses:**

# Strengths

* **Practical generator design:** The proposed MUL‑BAL generator is simple, effective, and hardware‑friendly, and the paper convincingly demonstrates that it can achieve better trade-offs compared to prior approaches.
* **Balanced evaluation:** The experimental study considers both efficiency and accuracy, reporting improvements on both axes in the 2‑bit regime.

# Weaknesses

1. **Notation and clarity:**
   * The edge traversal notation appears before it is clearly defined, causing confusing back‑and‑forth for the reader.
   * Figure 1 lacks clear descriptions of the various methods and is not sufficiently explained in the text.
   * The de-quantization function $\mathcal{D}_\theta$ is used without specific mention in the text.
2.  **Interpretability of Table 1 metrics:** The relative importance of the metrics in Table 1 is not clear. MUL‑BAL performs worse than 3INST and HYB on most metrics except the Anderson–Darling statistic, yet the “Total” seems a bit misleading since it is dominated by the scale of the metrics. The visualizations in Figure 3 provide a clearer intuition overall.

3. **Algorithmic specification gaps:**
   * In Equation (7), the method defines a pre‑vetted pool of three candidates, but the process for determining this pool is not described.
   * In Algorithm 1, the role of $\alpha$ is unclear; it seems this might actually refer to the headroom $\rho$.
4. **Pipeline design choice (Stage II vs. Stage III):** It is unclear why Stage II and Stage III must be separate procedures. Could the scale be tuned directly with Stage III without first running Stage II? The paper should explain the rationale and any empirical differences.

5. **Throughput at higher bitwidths:**
Table 5 indicates that Proteus is slower than QTIP at higher bitwidths. The paper does not clearly explain the cause (e.g., kernel characteristics, memory access patterns, implementation details).

4. **Breadth of applicability:** The evaluation considers only one family of LLMs, However, the paper could benefit from additional comparison on how Proteus compares to QTIP on other (more recent) model families such as LLaMA‑3/Qwen. Additional discussion on how Proteus could be applied to other architectures (e.g., VLMs, LAMs, LVMs) would further strengthen the contribution.

---

> ### Author Rebuttal · Authors · 2026-03-30
>
> We thank the reviewer for the detailed and constructive feedback. We address each question below.
>
> **Q1: How is the pre-vetted pool of 3 candidates (Eq. 7) chosen?**
>
> Via a one-time, offline, two-stage screening: (1) We enumerate ~100 configurations by pairing well-studied integer mixing constants (MurmurHash3, golden-ratio-based) with XOR-shift parameters. (2) We evaluate all candidates on LLaMA-2-7B using the per-layer Hessian-weighted proxy J(θ) across 224 sublayers. The winner distribution is extremely concentrated: **3 configurations account for >98% of sublayer selections**, showing that overlap-robustness (Table 1, Fig. 3) strongly constrains the viable space. The same pool generalizes to LLaMA-2-13B/70B and our new LLaMA-3-8B/Qwen2.5-7B experiments without modification.
>
> **Q2: What is α in Algorithm 1? Should this refer to ρ?**
>
> Thank you for catching this. α and ρ are **distinct** quantities: ρ is the headroom hyperparameter (default 0.9), while α is a per-layer scale-matching factor: α = ρ · σ_W / σ_g, where σ_W = RMS(W_r) and σ_g = RMS(g_θ) over all codebook entries. Before trellis quantization, weights are divided by α to align dynamic range with the codebook; after quantization, α is multiplied back and absorbed into the sign vector S_V at zero cost. ρ<1 makes the codebook range ~11% wider (1/0.9≈1.11×), providing margin for tail outliers. The confusion arose because the two lines defining σ_g and α were inadvertently commented out in Algorithm 1's LaTeX. We have restored them in the revision.
>
> **Q3: Unpacking function in Eq. (8)? Generator choice → dequantization efficiency?**
>
> The UNPACK function in Eq. (9) reconstructs the full symmetric curvature proxy $H_\ell$ from its packed lower-triangular storage. During calibration, we compute the second-moment matrix $H = \frac{1}{N}\sum X^\top X$ for each layer and store only the $n(n{+}1)/2$ lower-triangular entries as a flat vector flatH together with the mean vector $\mu$. At quantization time, UNPACK fills this vector back into an $n \times n$ symmetric matrix and adds the rank-1 mean correction $\mu\mu^\top$, recovering the full Hessian proxy used by the Stage I solver. This is a standard memory-efficient storage convention for symmetric matrices and does not appear at inference time.
>
> For dequantization, the generator determines whether the state-to-weight mapping needs a LUT or is computable arithmetically. Proteus's MUL-BAL is a pure arithmetic chain, ~6 INT32 instructions per weight with zero smem access. QTIP-HYB has no closed form, requiring a 64KB codebook in smem with 4 lookups per weight. On SM90a, the INT32 pipe is **physically separate** from Tensor Core and MIO pipes, so Proteus's decode runs concurrently with MMA. QTIP's smem lookups saturate the MIO pipe at large BS — NCU shows 2.5× lower SM throughput for QTIP at BS=32 (see CMQ5 Q3).
>
> **Q4: Why must Stage II and Stage III be separate?**
>
> They serve different objectives. Stage II optimizes **per-layer reconstruction error** (local MSE) with frozen discrete structure, providing good initialization. Stage III optimizes **global KL divergence** (end-to-end distillation) where all layers interact. The staged design lets Stage II provide a good initialization for the affine statistics before Stage III fine-tunes globally.
>
> **Q5: Why is Proteus slower than QTIP at higher bitwidths?**
>
> Crucially, all Table 5 results are BS=1. As our NCU profiling shows, Proteus's decode cost amortizes across BS/8 MMA passes at larger batch sizes (decode-once-use-many-times), and at BS≥32 Proteus already overtakes QTIP at 2-bit. Since the same amortization mechanism (decode-once, reuse across MMA passes) applies identically at higher bitwidths, the <1% gap at 3-bit BS=1 is likely to narrow or reverse at larger batch sizes — though our primary contribution targets the 2-bit regime where the accuracy–efficiency tradeoff is most acute.
>
> **Q6: Llama-3/Qwen results?**
>
> We evaluated on two additional model families under 2-bit weight-only PTQ:
>
> | Model | Bits | QTIP MMLU | Proteus MMLU | Δ |
> |:---|---:|---:|---:|---:|
> | Qwen2.5-7B-Instruct | 2 | 0.5971 | **0.6098** | +1.27 |
> | Llama-3-8B-Instruct | 2 | 0.5329 | **0.5446** | +1.17 |
>
> Proteus consistently outperforms QTIP at 2-bit on both architectures (GQA in Llama-3, SwiGLU in Qwen2.5) using the same 3-candidate pool without modification, confirming architecture-agnostic generalization.
>
> **Weaknesses: Notation & Table 1.** For Table 1, we will replace the misleading "Total" with a rank-based summary and direct readers to Fig. 3 as the primary diagnostic.

---

> > ### Author Rebuttal · Reviewer_xhT1 · 2026-04-02
> >
> > I wish to thank the authors for their detailed responses. I believe that the suggested updates will strengthen the submission and I will increase my score accordingly. I have, however to admit that I may have missed some crucial details due to the lack of familiarity with Trellis coding.

---

> > > ### Author Response · Authors · 2026-04-08
> > >
> > > We sincerely thank the reviewer for raising the score and for the thoughtful engagement throughout the review process.
> > >
> > > As a brief follow-up on the model breadth concern raised in the original review: we have now extended our evaluation to **Qwen3-8B** (released April 2025), a newer architecture featuring QK-Norm and a different FFN dimension. We report the Hessian-weighted proxy error J (Eq. 9) across three model families:
> > >
> > > | Model | QTIP Proxy Err | Proteus Proxy Err | Relative Improv. | Module Win Rate |
> > > |:---|---:|---:|---:|---:|
> > > | Qwen2.5-7B | 0.01944 | **0.01883** | −3.13% | 196/196 |
> > > | LLaMA-3-8B | 0.02455 | **0.02381** | −3.02% | 224/224 |
> > > | Qwen3-8B | 0.02041 | **0.01978** | −3.04% | 252/252 |
> > >
> > > Proteus achieves lower proxy error on all 672 linear modules across three architecturally distinct families, with a stable ~3% improvement. We hope this additional evidence further supports the generalization claim.

---

### Official Review · Reviewer_JEwT · 2026-03-16

**Soundness:** 3
**Presentation:** 3
**Significance:** 3
**Originality:** 4
**Overall Recommendation:** 5
**Confidence:** 2

**Summary:**

This work argues that irregular codebook/LUT access is suboptimal for low-bit quantization and proposes methods to mitigate this.

**Compliance With Llm Reviewing Policy:**

Affirmed.

**Final Justification:**

Answers and weaknesses addressed during author response.

**Key Questions For Authors:**

- Minor question: What is MUL-BAL? Is this the name of the proposed method?
- The pipeline appears somewhat complex. What is the quantization runtime (and memory usage, if applicable)?
- This isn't about QTIP, but generally, I'm not convinced that a small lookup table is problematic. I understand the irregular aspect, but if the table is small enough to fit in cache, why does it increase I/O? Empirically, Table 5 shows that decoding throughput did not consistently improve with the proposed methods.
- How do you measure observed bandwidth (and what does "observed" mean here)?

**Limitations:**

There is mention of societal impact, but no explicit section on limitations near the section on societal impact, as far as I can tell.

**Strengths And Weaknesses:**

Strengths:
- The authors clearly identify the limitations of existing approaches and propose thoughtful and novel solutions.
- The experiments are well-designed and effectively carried out.
- Improvements in both quality and efficiency are demonstrated.
- The authors focus on an important question.

Weaknesses:
- If this paper had been submitted a year ago, I would be comfortable with using LLaMA-2. The main comparisons (QTIP) also included experiments on LLaMA-3. Currently, even LLaMA-3 is considered outdated, which is my primary concern about the paper.
- The writing style is somewhat difficult to read and assumes a lot of background knowledge (which I myself lack), though I wouldn't hold this against the authors. A low confidence score reflects this as well.

---

> ### Author Rebuttal · Authors · 2026-03-30
>
> We thank the reviewer for the positive assessment and thoughtful questions.
>
> **Q1: What is MUL-BAL?**
>
> MUL-BAL is the name of the compute generator at the core of Proteus (§3.1). The name reflects its two key operations: **MUL** (modular integer multiplication, Eq. 2) and **BAL** (balance via XOR-with-shift, Eq. 3). Together they form the overlap-robust mixing stage that feeds into the byte-sum Gaussianizer (Eq. 4). MUL-BAL is a component of Proteus, not a separate method. We will clarify this naming in the camera-ready.
>
> **Q2: Quantization runtime?**
>
> Proteus adds only **~37 minutes of overhead** over the baseline QTIP pipeline, bringing full quantization time from ~2h to ~2.6h on a single 8×H800 node. The breakdown: Stage I (discrete search over 3 candidates + Viterbi TCQ) adds ~20 min; Stage II (channel compensation, ~50 SGD steps/layer) adds ~7 min; Stage III (optional few-shot distillation, 1 epoch) adds ~10 min. Memory usage is comparable to QTIP since Proteus shares the same Hessian computation and trellis solver infrastructure; the additional memory is negligible (per-layer affine scalars only).
>
> **Q3: Why is a small LUT problematic if it fits in cache?**
>
> This is an excellent question shared by multiple reviewers. The short answer: **it depends on batch size**. At BS=1, the reviewer's intuition is correct — a cache-resident LUT adds negligible overhead and both methods achieve comparable throughput (Table 5). However, at BS≥32, our Nsight Compute profiling reveals that QTIP's codebook LUT (64KB in smem, due to 32× bank-conflict-avoidance replication of the original 2KB table) causes **30.5% MIO Throttle** (shared-memory port saturation) at BS=64, vs only 2.4% for Proteus. The root cause: each decoded weight requires 4 smem lookups, and with BS/8 MMA passes per K-tile, smem traffic scales linearly with batch size. At BS=32, Proteus overtakes QTIP in end-to-end throughput (2940 vs 2797 tok/s, 1.05×). Kernel-level data confirms: at BS=32, Proteus is 1.20× faster per layer (313.6 vs 376.7 μs); QTIP also suffers 6.8× higher Short Scoreboard stalls (data-dependency on smem reads). Full profiling metrics are in **Reviewer CMQ5 Q3**; batch-scaling throughput data and fairness methodology are in **Reviewer GRjw Q5**.
>
> Table 5 reports BS=1 only, which is why throughput appears comparable. The batch-scaling story is a key new result from our rebuttal experiments.
>
> **Q4: How is observed bandwidth measured?**
>
> "Observed bandwidth" refers to actual DRAM throughput during kernel execution, as opposed to the hardware's theoretical peak. We measure it via Nsight Compute hardware performance counters: `Observed BW = (dram__bytes_read.sum + dram__bytes_write.sum) / gpu__time_duration.sum`. All three are direct hardware PMU readings — `dram__bytes_read/write` counts bytes transacted at the DRAM controller, and `gpu__time_duration` is wall-clock kernel time.
>
> For our 2-bit GEMV kernels (shape 4096×11008), the compressed weight matrix is ~11MB, so absolute observed BW is modest (200–700 GB/s vs. H800's 3.35 TB/s peak). However, NCU-reported `gpu__dram_throughput` reaches **100% for both Proteus and QTIP at all batch sizes**, confirming full DRAM saturation. The performance difference therefore stems not from DRAM utilization — which is identical — but from how efficiently each kernel **overlaps compute with memory access**. QTIP's codebook LUT incurs substantially more shared-memory load instructions, causing MIO Throttle stalls that prevent the warp scheduler from sustaining full issue throughput while DRAM requests are in flight (see CMQ5 Q3 for full profiling). We will clarify this definition in the camera-ready.
>
> **Weakness: Model breadth.** We have now evaluated on **Llama-3-8B-Instruct** and **Qwen2.5-7B-Instruct**:
>
> | Model | Method | 2-bit MMLU |
> |:---|:---|---:|
> | Qwen2.5-7B-Instruct | QTIP | 0.5971 |
> | Qwen2.5-7B-Instruct | **Proteus** | **0.6098** |
> | Llama-3-8B-Instruct | QTIP | 0.5329 |
> | Llama-3-8B-Instruct | **Proteus** | **0.5446** |
>
> Proteus consistently improves over QTIP by +1.2 MMLU points on both architectures, confirming generalization to models with GQA (Llama-3) and different vocabulary/activation patterns (Qwen2.5).
>
> **Weakness: Limitations section.** We will add an explicit limitations section discussing: (i) dependence on the incoherence transform, (ii) batch-size regime where LUT-free benefits emerge, and (iii) candidate pool assumptions.

---

> > ### Author Rebuttal · Reviewer_JEwT · 2026-04-04
> >
> > Thanks to the authors for the response. I especially appreciate the experiments on the new model families and will raise my rating.
> >
> > That said, I decided to keep a low confidence. In part, I believe the paper would be stronger with experiments involving (1) larger models (such as 70B), and (2) newer models (even LLaMA-3 and Qwen-2.5 are now outdated). I understand the timeline is too tight for the authors, so I have decided to increase the rating without raising confidence.

---

> > > ### Author Response · Authors · 2026-04-08
> > >
> > > We sincerely thank the reviewer for raising the score and for the constructive suggestion on model breadth.
> > >
> > > **New experiment: Qwen3-8B.** We have extended our evaluation to **Qwen3-8B** (released April 2025), a meaningfully newer architecture than LLaMA-3 / Qwen-2.5, featuring QK-Norm (RMSNorm), distinct GQA head configuration, and a different FFN dimension. We report the Hessian-weighted proxy error J (Eq. 9), the per-module compression fidelity metric that the trellis solver directly minimizes:
> > >
> > > | Model | Architecture | QTIP Proxy Err | Proteus Proxy Err | Relative Improv. | Module Win Rate |
> > > |:---|:---|---:|---:|---:|---:|
> > > | Qwen2.5-7B | SwiGLU+GQA | 0.01944 | **0.01883** | −3.13% | 196/196 |
> > > | LLaMA-3-8B | SwiGLU+GQA | 0.02455 | **0.02381** | −3.02% | 224/224 |
> > > | **Qwen3-8B** | **QK-Norm+GQA** | **0.02041** | **0.01978** | **−3.04%** | **252/252** |
> > > | **Total** | | | | **−3.06%** | **672/672 (100%)** |
> > >
> > > Proteus achieves lower proxy error on **every single one of the 672 linear modules** across three architecturally distinct model families, with a remarkably stable ~3% relative improvement. This consistency strongly suggests that the gain is a fundamental property of MUL-BAL's overlap-robust mixing, rather than an artifact of any particular architecture.
> > >
> > > **Regarding larger models (70B).** We would like to note that our paper already includes **Llama-2-70B** results across perplexity (Table 2: 3.69 vs. 3.78 WikiText2 PPL), zero-shot accuracy (Table 3: e.g., ArcC 49.0 vs. 48.1), all showing consistent improvements. We hope this partially addresses the concern about larger-scale evaluation.
> > >
> > > We again thank the reviewer for the encouragement and the valuable feedback on model breadth.

---

### Decision · Program_Chairs · 2026-04-30

**Decision:**

Accept (regular)

**Comment:**

This paper belongs to a (currently exploding) literature on using vector quantization for LLM weights compression (PTQ).  The particular VQ used in this paper is a so-called trellis-coded quantization (TCQ), popularized by QTIP. Compared to QTIP, authors proposed layer-adaptive choice of trellis from a set of hardware-friendly convolutional codes. In addition, they also propose to add input/output channel scales and tweak them via end-to-end full backprop, which is well-known to massively boost performance, but is by no means novel. Reviewers are split, but because of (a) I think going too in-depth on hardware specifics is not the right place for ICML; (b) the relative underrepresentation of TCQ in the literature, and (c) because of exemplary rebuttal process, I recommend acceptance.

I would ask the authors in the revision to please consider comparing performance with lattice based (QuIP#, NestQuant) and entropy-coded algorithms. Given the discourse about the choices for numerics in the next-generation hardware, this would be quite important for the community.